# Mechanisms of Action of the Peptide Toxins Targeting Human and Rodent Acid-Sensing Ion Channels and Relevance to Their In Vivo Analgesic Effects

**DOI:** 10.3390/toxins14100709

**Published:** 2022-10-17

**Authors:** Clément Verkest, Miguel Salinas, Sylvie Diochot, Emmanuel Deval, Eric Lingueglia, Anne Baron

**Affiliations:** 1CNRS (Centre National de la Recherche Scientifique), IPMC (Institut de Pharmacologie Moléculaire et Cellulaire), LabEx ICST (Laboratory of Excellence in Ion Channel Science and Therapeutics), FHU InovPain (Fédération Hospitalo-Universitaire “Innovative Solutions in Refractory Chronic Pain”), Université Côte d’Azur, 660 Route des Lucioles, Sophia-Antipolis, 06560 Nice, France; 2Department of Anesthesiology, University Medical Center Hamburg-Eppendorf, 20251 Hamburg, Germany

**Keywords:** ASIC, sodium channels, toxins, peptide, PcTx1, APETx2, MitTx, mambalgin, pain, nociception

## Abstract

Acid-sensing ion channels (ASICs) are voltage-independent H^+^-gated cation channels largely expressed in the nervous system of rodents and humans. At least six isoforms (ASIC1a, 1b, 2a, 2b, 3 and 4) associate into homotrimers or heterotrimers to form functional channels with highly pH-dependent gating properties. This review provides an update on the pharmacological profiles of animal peptide toxins targeting ASICs, including PcTx1 from tarantula and related spider toxins, APETx2 and APETx-like peptides from sea anemone, and mambalgin from snake, as well as the dimeric protein snake toxin MitTx that have all been instrumental to understanding the structure and the pH-dependent gating of rodent and human cloned ASICs and to study the physiological and pathological roles of native ASICs in vitro and in vivo. ASICs are expressed all along the pain pathways and the pharmacological data clearly support a role for these channels in pain. ASIC-targeting peptide toxins interfere with ASIC gating by complex and pH-dependent mechanisms sometimes leading to opposite effects. However, these dual pH-dependent effects of ASIC-inhibiting toxins (PcTx1, mambalgin and APETx2) are fully compatible with, and even support, their analgesic effects in vivo, both in the central and the peripheral nervous system, as well as potential effects in humans.

Early work by Krishtal et al. in 1980 demonstrated for the first time that application of extracellular acid could evoke inward currents in sensory neurons [1]. Later on, they were attributed to Acid-Sensing Ion Channels (ASICs) [2,3], which are members of the epithelial Na^+^ channel (ENaC) and Degenerin (DEG) ion channel superfamily [4]. Understanding the proton-dependent activation and modulation of ASICs as well as their pathophysiological roles was the subject of active research since their molecular identification and cloning in the late 1990s. Significant efforts were made to discover pharmacological tools to help decipher their function, and peptides isolated from venoms turned out to be a powerful resource for it. Tissue acidosis is associated with pain and ASICs emerged as major pH sensors in sensory neurons where protons also directly affect other receptors such as the Transient Receptor Potential (TRP) vanilloid 1 TRPV1 [5,6,7], some members of the two-pore domain potassium channel family [8], cyclic nucleotide gated (CGN) channels [9,10] and G-protein coupled receptors [11].

Here, we discuss recent advances in our understanding of ASIC mechanisms of activation by acidic pH and how peptide toxins exert their complex molecular effects on them, with a special emphasis on the in vivo consequences on pain and their use both as pharmacological tools and potential analgesic compounds.

## 1. Molecular and Functional Properties of ASICs

### 1.1. Subunits Diversity and Structure

Functional ASICs are formed by the homo- or heterotrimeric association of identical or homologous subunits [12,13,14] (Figure 1A), each subunit comprising more than 500 amino acids and two transmembrane domains, a large extracellular loop, and intracellular N- and C-termini with a re-entrant N-terminus loop (Figure 1B,C).

Four genes (ACCN1 to ACCN4) encode at least six different ASIC subunits (Table 1) sharing more than 50% amino acid identity: ASIC1a, ASIC1b, ASIC2a, ASIC2b, ASIC3, ASIC4 (ASIC5, also named BLINaC/BASIC and coded by the ACCN5 gene, only shares 30% amino acid identity and cannot be considered as a genuine ASIC subunit). The difference between the a and b variants of ASIC1 and ASIC2 relies on the first N-terminal third of the subunit (Figure 1B), including the cytoplasmic N-terminal domain, the re-entrant loop (forming part of the pore with the HG motif), the first transmembrane domain TM1 (forearm and wrist domains), and part of the extracellular loop (the palm β1 and β3 sheets, the β-ball β2 sheet and the entire finger domain).

The first crystal structure of an ASIC was solved in 2007 by the group of E. Gouaux from cASIC1 (chicken ASIC1), the chicken ortholog of rat ASIC1a (rASIC1a) [12]. Each subunit was represented as a hand holding a ball and divided into finger, thumb, palm, knuckle, β-ball, wrist, and forearm (transmembrane domains) domains (Figure 1C). An “acidic pocket” containing several pairs of acidic amino acids is present at the interface of each subunit and was proposed to be one of the pH sensors of the channel, whereas cations may access the ion channel by lateral fenestrations, then moving into a broad extracellular vestibule just above the inactivation gate and the selectivity filter (i.e., the structural element in the narrowest part of the pore that determines ionic selectivity) (Figure 1D) [12,15]. The most noticeable structural difference between human (h) ASIC1a (hASIC1a) and cASIC1 is a longer loop that extends down from the α4-helix to the tip of the thumb, due to two extra amino acids (D298 and L299) absent in all other ASIC isoforms [16].

The lowest amino acid identity is 52% between cASIC1 and rASIC3, while rat and human ASIC orthologs show amino acid identities between 83.68% (for ASIC3) and 99.02% (for ASIC2a) (Table 1). Interestingly, hASIC3, but not rASIC3, has three splice variants (a, b, c), resulting in differences in the C-terminal domain, hASIC3a mRNA being the main isoform expressed in human neuronal tissues, although hASIC3c was also significantly detected. A higher level of sequence variability for the same isoform is observed between hASIC3a and rASIC3 or mouse ASIC3 (mASIC3) orthologs. While experimentally determined structures are still lacking for ASIC2 and ASIC3, recent major advances in structure prediction using machine learning have allowed the generation of models for those ASICs, shedding a new light on potential structural variations underlying the functional differences between ASICs [17].

### 1.2. pH-Dependency

Homo- or heterotrimeric cloned ASICs were found to be voltage-insensitive but highly pH-sensitive upon heterologous expression in *Xenopus* oocytes or in mammalian cell lines. They are sodium selective, with additional low calcium permeability for ASIC1a and hASIC1b [3,18,19]. They are activated by a fast-extracellular acidosis from conditioning physiological pH to acidic test pH and inactivated by sustained extracellular acidosis. Interestingly, rat and human ASIC3 channels can be also activated at neutral (7.4) pH by lipids (arachidonic acid and lysophosphatidylcholine) [20,21] and hASIC3a channels have been shown to be sensitive to both acidic and alkaline pH [22].

The ASIC2b and ASIC4 subunits do not form functional proton-gated channels by themselves, but ASIC2b can associate with other ASIC subunits to confer new properties and regulations to heterotrimeric channels [13,23,24]. ASIC currents are generally transient even if the acidification is maintained, but a sustained phase is associated with expression of ASIC3 or with the presence of the ASIC2b subunit in heterotrimers (Figure 2). A sustained plateau phase is also associated with hASIC1b current, but not with rASIC1b [19]. Two types of sustained currents have been described for ASIC3: a window current at pH around 7.0 (Figure 2) resulting from the overlap of pH-dependent activation and desensitization curves, and a sustained current induced by more acidic test pHs. The TM1 domain modulates the pH-dependent activation, thus contributing to the window current near physiological pH and, combined with the N-terminal domain, the TM1 domain is also the key structural element generating the low pH-evoked sustained current [25].

ASICs differ in their biophysical properties depending on their subunit composition, notably in their desensitization time constants. rASIC3, rASIC1a and rASIC1b show significantly faster desensitization kinetics than rASIC2a (Figure 2). Experiments were conducted on heterologously expressed cloned channels by co-expressing two, or more rarely three (or concatemers), different subunits. The stochiometry of association in heterotrimers (i.e., the relative number of each subunit in the channel) is therefore hard to precisely determine in these conditions. The ASIC1a and ASIC2a stoichiometry was at least investigated, showing no preferential association [26].

ASICs differ in their pH sensitivity, and the functional diversity obtained by combining the different subunits in homo- or heterotrimers (Table 2, with references in legend) allows these channels to detect a wide range of pH changes between pH 7.2 and pH 4.0. Sigmoidal curve fit of pH-dependent activation is used to determine the test pH_0.5_, inducing the half-maximal activation that can vary between 6.8–6.3 (rASIC3) and 5.0–3.8 (rASIC2a). ASICs are desensitized depending on the conditioning pH, which is represented by the pH-dependent sigmoidal curve of steady-state desensitization (SSD) and by the conditioning pH_0.5_ of half-maximal SSD that can vary between 7.4–6.8 for rASIC1a and rASIC3 and 6.0–4.7 for hASIC2a (Table 2). These values are generally less acidic than the ones for activation, which means that the SSD mechanism in the presence of a sustained extracellular pH acidification will highly influence the amplitude of an ASIC current triggered by a subsequent rapid drop in pH. The sustained ASIC3 current results from incomplete inactivation and is activated at test pH 6.0 and below for hASIC3a and at pH 6.5 and below for rASIC3.

### 1.3. pH-Dependent Gating

The molecular mechanism of pH-dependent gating of ASICs was studied through combined approaches including electrophysiology, mutagenesis, molecular dynamic simulations, X-ray crystallography and cryo-electron microscopy (cryo-EM), along with the pharmacological use of ASIC-targeting animal toxins.

The structures of the three conformational states involved in H^+^-dependent gating of homotrimeric cASIC1 were solved by X-ray crystallography and Cryo-EM: resting state [50] (Figure 3A), open state [51] (Figure 3B) and desensitized state [12] (Figure 3C). Cryo-EM structures of the hASIC1a in its closed state have also been solved, in complex with the toxin mambalgin-1 [52], and in complex with a specific nanobody Nb.C1 [16]. At the interface of each subunit of the trimeric channel, an acidic pocket is formed by intra-subunit contacts between the thumb, the β-ball and the finger domains, together with residues from the palm domain on the adjacent subunit (Figure 1A,D and Figure 3) [12,50].

Upon activation by extracellular acidic pH, protonation of the acidic pocket (Figure 3A,B) leads to its collapsed conformation, which is stabilized by the formation of three pairs of carboxyl–carboxylate interactions between the side chains of aspartate and glutamate residues [12,50]. Cl^−^ ion may play a role in channel gating by stabilizing the collapsed conformation of the acidic pocket at low pH, which seems state dependent since this bound Cl^−^ is absent in the resting state at high pH [53]. The motion of thumb helices α4/α5, resulting from collapse of the acidic pocket, induces a global motion of the thumb domain, which is directly connected to the transmembrane domain through a non-covalent contact forming a part of the wrist region [12,54]. In parallel, anchoring of the α5 helix against the palm of the adjacent subunit induces bending of the lower palm (β1 and β12 strands) toward the transmembrane domains (TM1 and TM2) to which they are covalently connected to form the other part of the wrist region. All together, thumb and lower palm domain motions lead to a rotation of each subunit around the scaffold formed by the knuckle and upper palm domains [50] that induces a translation of TM1 and TM2 leading to the expansion of extracellular fenestrations and to an iris-like opening of the channel gate (Figure 3B).

Ions are then enabled to pass through the selectivity filter of the pore (GAS belt motif, between TM2a and TM2b and HG motif in N-terminal re-entrant loop; Figure 1C) [15,51,55]. The Lys212 of the palm domain is deeply anchored to the thumb domain of the adjacent subunit and seems critical to facilitate the cooperativity between subunits during the global rotation of the extracellular domains of all the subunits [12,15,51,56]. In the lower palm domain, an inter-subunit hydrogen-bond network, close to the wrist region, seems also critical for the correct propagation of conformational changes leading to the expansion of the extracellular fenestration [57,58] (Figure 3B). Several residues at the extracellular side of the transmembrane domain that form contacts within each subunit in desensitized and resting state [12,39] are disrupted after the iris-like opening of the pore [50]. It is interesting to note the arginine in this region that seems also necessary to mediate the potentiation of ASIC currents by lipids [59]. In addition to acid-induced activation, hASIC3 is also sensitive to alkalization, and this property is supported by two arginine residues only present in the human channel and also located close to the boundary between the plasma membrane and the extracellular medium [22].

During prolonged acidification, the β11-β12 linker that demarcates the upper and lower palm domains undergoes a substantial conformational change induced by the switch in sidechain orientations of two residues [60,61,62]. This plays the role of “molecular clutch” allowing transmembrane domains to relax back into a “resting-like” conformation to permit rapid desensitization by uncoupling the conformational change of the upper extracellular domain from the lower part of the channel leading to the narrowing of the fenestration and the closing of the inactivation gate (Figure 3C) [50,61,62,63,64,65]. Just under the “molecular clutch”, a fourth pair of carboxyl–carboxylate interaction between the side chains of glutamate residues was identified [12] that probably influences its stability [66,67]. Moreover, the previously mentioned Lys212, in a loop immediately above the β11-β12 linker, binds a Cl^−^ anion located in the thumb domain of an adjacent subunit [12,50,53] and could explain the mutations in the thumb domain also influencing desensitization [64,68,69,70].

Finally, when returning to physiological pH, the channel would return to the resting state after deprotonation of acidic residues that drive the expansion of the acidic pocket, allowing the β11-β12 linker to revert back to a non-swapped conformation (Figure 3A) [50,61].

The intra- and inter-subunit network of H-bonds, salt bridges and carboxyl–carboxylate pairs involving several residues with different pKa values, highlights the complexity of the pH-dependent gating of ASICs and explains the different pH-dependent activation and desensitization characteristics of the various homo- or heterotrimeric channels in different species (Table 2), since several domains are directly or indirectly involved in both mechanisms via their intra- and inter-subunit connections.

Among the channels and receptors that respond to acidic pHs, only the proton-activated chloride channel (PAC) is, like ASICs, directly activated by protons through a complex and dedicated mechanism [71,72]. Interestingly, this channel also has a trimeric structure with a large extracellular domain, comprising fenestrations and acidic pockets, whose conformation change is transmitted to transmembrane domains after protonation. This suggests a complex convergent evolutionary process to achieve the pH sensing property of these two unrelated channels, albeit with completely different mechanisms at the molecular level. Very recently, a lysosomal proton (and K^+^) channel [73] with unrelated sequence and structure with ASICs was shown to be also activated by protons via a still unknown molecular mechanism. Proton-mediated gating of the capsaicin receptor TRPV1 is dependent of one key residue of the pore region (Phe660) [7], the inactivation of CNG channels is controlled by extracellular protons leading to the collapse of the pore via the titration of a single glutamate residue within the selectivity filter [10], and the two pore domain potassium channels (K_2_P) are modulated by protons through the titration of a key residue in the pore [8].

### 1.4. Pathophysiological Roles in Pain Sensing

Relying on their pH-dependent gating properties, ASICs were involved in several functions associated with physiological and/or pathological extracellular pH variations [4,74,75,76,77,78,79,80,81]. We will focus here on pain sensing. Besides metabolic disorders producing systemic pH changes, there are other pathophysiological conditions that result in local pH variations, generally associated with increased pain perception. During pathophysiological conditions like inflammation, tissue injury, ischemia or cancer, extracellular pH can drop from physiological values (generally around 7.4) to values around 6.5 or even below. In rodent models, local extracellular pH can for instance decrease to 6.8–6.0 in implanted tumor, to 5.8 in carcinoma, to 6.0–4.0 in bone cancer, to 6.8 upon inflammation, to 5.7 in arthritis, to 6.5 in muscle after incision, to 6.9 during heart angina, to 6.9 in joints in osteoarthritis or rheumatoid arthritis, and to 6.5–6.2 in mouse brain after traumatic injury or stroke [82,83,84,85]. In humans, local extracellular pH was found to decrease to 6.0–5.4 in abscess, to 5.7–5.4 in human malignant tumors, to 7.0–6.0 in joint synovial fluid of osteoarthritis patients, to 6.9 in gout, to 6.4 in melanoma, or to 6.7 intracutaneously after muscle exercise [81,84], and localized skin tissue acidification (pH ≥ 6.0) causes pain in humans [86,87].

Tissue acidosis occurs therefore in a variety of pathological painful conditions, and ASIC subtypes expressed in nociceptive neurons have all the hallmarks of pain sensors. Several reviews have summarized the role of ASICs in the peripheral nervous system in nociception and also in proprioception [74,75,76,78,79,80,88]. ASICs are highly sensitive to moderate acidifications, being for instance 10-fold more sensitive than the heat, capsaicin and proton-sensitive channel TRPV1 also expressed in peripheral sensory neurons. ASICs can generate sustained depolarizing currents upon prolonged tissue acidification compatible with the detection of non-adapting pain. Regulation of their activity by several pain-related mediators beside protons (inflammatory factors, neuropeptides, lipids, etc.) [21,89,90] led to the notion of coincidence detectors, especially for ASIC3, associated with pain detection and peripheral sensitization processes in pathophysiological situations like inflammation or chronic pain [81,91].

It is important to note that ASICs in the pain pathways are expressed not only in sensory neurons but also in dorsal horn neurons of the spinal cord involved in pain processing as well as in the brain, where they could be involved in synaptic transmission and plasticity, activated by the acidification of the synaptic cleft after the co-release of the acidic content of neurotransmitter synaptic vesicles, in particular in the case of chronic pain situations leading to central sensitization processes. Homotrimeric ASIC1a and heterotrimeric ASIC1a/2a were found to be postsynaptic receptors activated in several brain structures in glutamatergic and GABAergic neurons, where they could generate 3–10% of the synaptic current, even 20% at GABAergic synapses, and be involved in diverse forms of synaptic plasticity [77,92]. In the central nervous system, presynaptic or postsynaptic ASICs have thus been proposed to modulate learning and memory and to play a role in epilepsy and mood disorders as well as in neuronal damages associated with stroke and Alzheimer’s disease [77,92,93].

## 2. Dual Effects of Animal Toxins Targeting ASICs

Development of the pharmacology of ASICs was very important for studying their structure, their molecular and cellular functions, and their pathophysiological roles, in combination with knockout or knockdown animals. The pharmacology of ASICs includes poorly selective modulators like synthetic amiloride, GMQ (2-guanidine-4-methylquinazoline), diminazene and non-steroidal anti-inflammatory drugs (NSAIDs), endogenous modulators like lipids, nitric oxide (NO), extracellular cations, polyamines such as agmatine or spermine, neuropeptides (dynorphin A, big dynorphin, RFamide-related peptides), natural compounds among which there are vegetal compounds particularly used in traditional Chinese medicine and finally much more selective animal peptide and protein toxins [94,95].

Four ASIC-targeting peptide toxins have been extensively characterized to date: the peptide toxins PcTx1 (psalmotoxin) from spider [96], APETx2 from sea anemone [97], and mambalgin (Mamb, with isoforms 1, 2 and 3) from mamba snakes [33,98], and the heterodimeric protein MitTx from coral snake [99].

The mechanisms by which toxins modulate ASIC gating are best described for ASIC1a and PcTx1 [29,31,32,47,60,100,101,102,103], ASIC1a and MitTx [51,99,104] and ASIC1a and Mamb [33,34,49,52,105,106,107,108] through combined approaches including electrophysiology, mutagenesis, molecular dynamics simulations, X-ray crystallography and cryo-EM, which allow to propose a mechanism for the pharmacological effects of these gating modifier toxins, as well as for APETx2 whose binding site on ASIC3 is not yet formally identified, which helps to understand their complex pH-dependent dual effects.

### 2.1. Preliminary Remarks on the Models and Data Interpretation

Depending on the subunit composition of ASICs, on the animal species and on extracellular pH variations, three different macroscopic effects could be observed on whole-cell currents flowing through heterologously expressed cloned ASICs when toxins are applied at physiological conditioning pH 7.4: inhibition (INH, Table 3 with references included) of the peak H^+^-gated current, potentiation (POT, Table 3) of the peak H^+^-gated current, or activation (ACT, Table 3) of the current without any change in pH.

These complex and sometimes opposite pH-dependent effects of ASIC-targeting peptide toxins could question the validity and/or extrapolation to humans of some in vivo data especially against pain. We will present evidence on how these dual effects of ASIC-targeting animal toxins on both rodent and human channels are fully compatible with, and even support, the analgesic effects seen in vivo in rodents and potential effects in humans.

The expression system should also be taken into account in interpreting these effects, as opposite results were sometimes observed on cloned ASICs whether they were expressed in *Xenopus* oocytes (🔾, Table 3) or in mammalian cells (◆, Table 3). For example, potentiating effects of APETx2 on rASIC1b and rASIC2a, as well as on rASIC1b/3 and rASIC2a/3 currents were reported from *Xenopus* oocytes, whereas similar concentrations of toxin were reported to have no effect on the same channels expressed in transfected mammalian cells. These effects observed in *Xenopus* oocytes should thus be considered with caution if they were not confirmed in transfected mammalian cells and/or in native currents recorded from neurons, particularly to extrapolate in vivo effects in mammals.

Furthermore, in structure-activity studies, the impact of a mutation on ASIC pH-dependent gating should be carefully controlled since the effect of mutations strongly affecting the pH-dependent gating can introduce a bias, making it difficult to draw a formal conclusion concerning the direct involvement of a channel residue into the binding site of a toxin.

### 2.2. PcTx1 and Related Toxins

#### 2.2.1. Pharmacological Profile

This peptide of forty amino acids was isolated from the South American tarantula *Psalmopoeus cambridgei* venom [96]. It folds according to the inhibitor cystine knot (ICK) motif [100,119]. Discovered as a potent inhibitor of cloned rASIC1a (IC_50_ = 0.3–3.7 nM), PcTx1 applied at conditioning physiological pH 7.4 was also shown to inhibit rASIC1a/3 at higher concentrations (25–100 nM) (Table 3, references included) and to inhibit hASIC1a at test pH 6 with an IC_50_ of 13 nM [31]. Inhibition of rASIC1a/2b was only reported once in the *Xenopus* oocytes expression system [40] but was not confirmed from mammalian cell experiments performed with up to 300 nM of PcTx1 (unpublished data). Depending on the pH conditions, PcTx1 also shows potentiating effects on rASIC1b, hASIC1a, hASIC1b, hASIC1a/2a and on cASIC1 currents (Table 3) [19,31,32] and can also directly activate cASIC1a (EC_50_~189 nM) [98,112].

PcTx1 was reported to exert complex state-dependent effects on ASIC1a- and ASIC1b-containing channels, which would depend on its concentration but also on the animal species, on the pH-dependent properties of the channels and on the pH at which the toxin is applied. For example, whereas exerting almost no effect on the hASIC1a current activated from conditioning pH 7.4 to test pH 5.0 [19,30,111], PcTx1 was reported to exert an inhibitory effect on the current maximally activated from conditioning pH 7.2, or on the current activated from conditioning pH 7.4 to test pH 6.0 at high concentration [30,31], and a potentiation of the current submaximally activated from conditioning pH 7.4 to test pH 7.2–6.2 [19] (Figure 4B). However, if inhibitory effects were obtained both from *Xenopus* oocytes and mammalian cells data, potentiating effects of PcTx1 on the rASIC1b, hASIC1a, hASIC1b and hASIC1a/2a need to be considered with caution because they are only obtained from *Xenopus* oocytes experiments.

PcTx1 has no effect on the other ASIC channels, nor on a variety of Kv, Nav and Cav channels [96]. The best specificity of the toxin for ASICs is supported by binding experiments with an iodinated form of the peptide, which shows similar binding properties on rat brain membranes and heterologously expressed cloned ASIC1a channels [47].

Despite these dual effects described on heterologously expressed cloned ASICs, the effects of PcTx1 on native neuronal rodent ASIC currents were mostly inhibitory [37,120,121], and generally used to support the participation of homotrimeric ASIC1a in the whole-cell ASIC current. The observed inhibition of native neuronal ASIC currents is in good agreement with in vivo analgesic effects confirmed by genetic invalidations, particularly after central (intrathecal, i.t., or intracerebroventricular, i.c.v.) injections of the peptide in different animal pain models (chemical, inflammatory and neuropathic pain) [122,123].

The spider peptide Hm3a (π-TRTX-Hm3a, from *Heteroscodra maculata*) shares high sequence similarity with PcTx1 (37 amino acids, with 5 different amino acids and three residues missing at the C-terminus) but is more resistant to enzymatic, chemical and thermal degradation although in vivo studies are yet to confirm this efficacy. It inhibits the rASIC1a current with an affinity (IC_50_ = 1.3–2.6 nM) similar to PcTx1, as well as hASIC1a (IC_50_ = 0.52 nM), and potentiates rASIC1b at higher concentration (EC_50_ = 46.5 nM) as well as rASIC1a/1b (EC_50_ = 17.4 nM) and hASIC1b (EC_50_ = 178 nM), all recorded from *Xenopus* oocytes [124].

Another spider toxin, Hi1a (from *Hadronyche infensa*), is a double knot peptide composed of 75 amino acids that looks like two PcTx1 in tandem (each with 62% and 50% identity with PcTx1) [125,126]. It inhibits rASIC1a (IC_50_ = 0.4 nM), as well as hASIC1a (IC_50_ = 0.5 nM) currents with an affinity similar to PcTx1, and potentiates rASIC1b at higher concentration (EC_50_ = 46 nM), with a more durable effect than PcTx1 on currents recorded from *Xenopus* oocytes [125]. Hi1a does not exert a major effect on a panel of human ion channels involved in cellular excitability (hNav1.5, hKv4.3/hKChIP2, hCav1.2, hKv11.1/hERG, hKv7.1/hKCNQ1 or hKir2.1 currents) [127].

#### 2.2.2. PcTx1, a Gating Modifier Stabilizing Open and Desensitized States

##### 
Biophysical Mechanisms


When co-applied at physiological conditioning pH 7.4, PcTx1 is able to bind to the closed state of ASIC1a and to induce a conformational change that is, however, not sufficient to directly open the channel in most cases [103]. By mimicking a local protonation of the acidic pocket, PcTx1 triggers its H^+^-dependent collapsed conformation thus promoting both open and desensitized states [51], as evidenced by the apparent greater affinity of protons (leftward shift) for both the activation and SSD curves of rASIC1a and hASIC1a (Figure 4A,B) [19,29,30,31,128]. The amplitude of this shift is dependent on the concentration of PcTx1 [31]. Except in the case of cASIC1 [98] and of chimeric channels on which PcTx1 behaves as a direct agonist at pH 7.4 [32], PcTx1 does not directly open ASIC1a, probably because other key residues must be protonated by a pH value lower than 7.4 to cause the pore to open completely.

PcTx1 inhibitory effect on rASIC1a current from physiological conditioning pH 7.4 to every test pH is mostly due to its pH-dependent SSD promoting effect, whereas no more inhibition was observed from pH 8 instead revealing a potentiation of the current at test pH values in the activation curve pH range (7.2–6.2), due to the opposite potentiating effect by a leftward shift of the activation curve (Figure 4A).

Rat and human ASIC1a differ by five residues in the thumb domain, which render the pH-dependent SSD of hASIC1a less sensitive to pH and thus prevent the potent inhibitory effect of PcTx1 at conditioning pH 7.4, revealing instead a current potentiation of hASIC1a through the shift of the pH-dependent activation that can be seen at test pH 7.2–6.2, i.e., in the activation pH range only [19] (Figure 4A,B). Inhibition of hASIC1a can only be restored if PcTx1 is applied with slightly acidic conditioning pH (pH 7.2 in the presence of 1 nM PcTx1 [31]), i.e., in the range of the effect of PcTx1 on the pH-dependent SSD (Figure 4B), or by increasing the PcTx1 concentration to 3–10 nM to further shift the SSD curve above pH 7.4 [31]. The PcTx1 effect on currents submaximally activated from conditioning pH < 7.4 will result from the sum of inhibiting and potentiating effects.

PcTx1 promotes opening of rASIC1b and hASIC1b by acidic drop from physiological resting pH 7.4 (EC_50_~100 nM) through a leftward shift of their activation curve towards less acidic pH (Figure 4C,D), with almost no effect on the SSD curve. Consequently, PcTx1 does not inhibit the current maximally activated from conditioning pH 7.4, and potentiates the current submaximally activated from pH 7.4 to test pH 6.8–5.8, in the activation curve pH range. The absence of PcTx1 effect on the pH-dependent SSD of rASIC1b comes from an alteration of the contact between the toxin and the divergent adjacent upper part of the palm domain of ASIC1b (Figure 5A) [32,47]. PcTx1 is also able to constitutively activate cASIC1 at resting pH 7.4 [112] and to potentiate the H^+^-activated cASIC1 current [98], presumably stabilizing only the open state of the channel like for ASIC1b.

##### 
Structural Mechanisms


PcTx1 primarily binds to the thumb domain through Trp7 and Trp24 that anchor the toxin to Phe351 of cASIC1, crucial for the specificity of PcTx1 [51,103] but also inserts an arginine-rich hairpin into the acidic pocket, making polar interactions mimicking local protonation of the acidic pocket (Figure 5A). Together, these polar and non-polar interactions link the finger, β-ball, and thumb domains of one subunit and the palm domain of the adjacent subunit [47,51] and lead to the collapsed conformation of the acidic pocket, which characterizes both the open and the desensitized states (Figure 3B,C). Three PcTx1 molecules thus bind at three equivalent sites on one homotrimeric ASIC1.

A series of chimeras realized between rASIC1a and rASIC1b or between rASIC1a and rASIC2a showed that the thumb, the β-ball and the palm domains of the adjacent subunit are crucial in explaining the difference in sensitivity of PcTx1 between the different ASIC isoforms. When ASIC1a residues of the β-ball were exchanged with the ones belonging to the PcTx1-insensitive ASIC2a (mutant 1a-RDQ190,258,259KQE) [49]), PcTx1 cannot inhibit the channel and rather induces a strong potentiation, in good agreement with the fact that Arg190 was involved in the interface with PcTx1 [51]. A similar effect is also observed when a part of the β-ball domain of ASIC1a was exchanged for the one of ASIC2a [47], showing the crucial role of the β-ball domain in the mechanism of PcTx1 inhibition of ASIC1a via the leftward shift of its pH-dependent SSD (Figure 4A,B).

When part of the palm domain of ASIC1a was exchanged with that of ASIC2a [49], PcTx1 also exerted a potentiating effect that cannot be attributed to an indirect effect via a change of the pH-dependence of SSD and can only be interpreted as a loss of contact between PcTx1 and the palm domain [47,49] preventing the modulation of the pH-dependent desensitization by PcTx1. The palm domains are different between rASIC1a and rASIC1b subunits, and interestingly, PcTx1 is also not able to inhibit the rASIC1b current by shifting its pH-dependent SSD curve [32] (Figure 4C), but rather induces a potentiating effect through the shift of the pH-dependent curve of activation. Accordingly, the introduction of part of the palm domain of rASIC1a into rASIC1b restored the inhibition by PcTx1 [47]. A similar mechanism could also take place in the case of hASIC1b [19] and cASIC1 [60,64,130], which all have palm domains divergent from ASIC1a and are all potentiated by PcTx1, and also explains why PcTx1 is able to open cASIC1 at pH 7.4 [98]. Moreover, an overlap between the effects of PcTx1 and GMQ was shown [131], which is known to act through the β11–β12 linker of the palm responsible for the desensitization (molecular clutch), highlighting the role of the palm domain to mediate the effect of PcTx1 on the desensitization process.

PcTx1 was reported to exert no inhibition (applied at pH 7.4, Table 3) on heterotrimeric ASIC1a/2a highly expressed in central neurons along with homotrimeric ASIC1a, and even a small potentiation on mASIC1a/2a when applied at conditioning pH 7.9 [40]. This can also be explained by the pH-dependent gating properties of these channels that show a half-maximal SSD around pH 6.8 [114,115], i.e., more acidic than rASIC1a, rendering the PcTx1-induced shift of the pH-dependent SSD curve not sufficient to desensitize ASIC1a/2a at conditioning pH 7.9, whereas the potentiation induced by the shift of the pH-dependent activation takes place. Accordingly, PcTx1 was able to inhibit heterotrimeric ASIC1a/2a when applied at the slightly acidic conditioning pH 7.0 [114,115].

##### 
PcTx1-Related Compounds


The peptide toxin **Hm3a** has five amino acid substitutions compared to PcTx1 and is three residues shorter at the C-terminus [124]. Only the R28K substitution is found in the active site, but has little apparent effect on its potency. Similar to PcTx1, Hm3a produces different effects on rASIC1a and rASIC1b that also depend on the palm domain of the adjacent subunit diverging between the two isoforms (especially Arg175 and Glu177 of rASIC1a corresponding to Cys and Gly residues in rASIC1b, respectively) [32,47].

The toxin **Hi1a** with two PcTx1-like peptides in tandem [125,126] inhibits rASIC1a and hASIC1a currents with a similar affinity and shifts (at 5 nM) the pH_0.5_ of activation from 6.13 to 6.01 for rASIC1a, and from 6.22 to 6.04 for hASIC1a, suggesting a stabilization of the closed state [125]. However, the pH_0.5_ of SSD is also shifted from 7.33 to 7.47 for rASIC1a and from 6.96 to 7.37 for hASIC1a, suggesting that Hi1a also promotes the desensitized state similarly to PcTx1. It seems that the Hi1a two-fold structure allows stabilization of the ASIC1a closed state as in the case of Mamb, in addition to promoting SSD like PcTx1 but weaker. This double effect, promoting desensitization and stabilization of the closed state, explains the inhibitory potency of Hi1a, which is not completely independent of conditioning pH as proposed by Chassagnon et al. [125]. When an alkaline conditioning pH is chosen to avoid the inhibitory effect taking place through the shift of the pH-dependent SSD curve, it reveals the effect generated by the shift of the pH-dependent activation curve, i.e., inhibition for Hi1a and potentiation for PcTx1.

The **C5b compound** was developed from the molecular knowledge of the binding of PcTx1 to ASIC1a. It binds in the acidic pocket, thus inhibiting hASIC1a and mASIC1a currents in a pH-dependent manner with an affinity decreasing with the acidification of the test pH value, as expected for a competitive proton inhibitor (IC_50_ = 22 nM, 100 nM or 7 µM when hASIC1a current is elicited from a conditioning pH 7.4 to test pH 6.7, 6.0 or 5.0, respectively) [132,133]. It is therefore more potent at mild than at extreme test pHs. C5b shifts the pH-dependent activation of hASIC1a towards lower pH, with a pH_0.5_ value shifted from 6.57 to 6.4 by 100 nM C5b, and it reduces the maximal (pH 5.0-evoked) current, in good agreement with a mechanism of competition between C5b and protons, further suggesting that C5b prevents the collapse of the acidic pocket, leading to a stabilization of the closed state. On mouse brain slices, C5b (100 nM) is able to inhibit the ASIC part of EPSCs recorded in the anterior cingulate cortex as well as the LTP induction in the hippocampal CA3–CA1 pathway [132,133], and when it is i.v. injected in mice, C5b appears to cross the blood–brain barrier [133]. However, the C5b compound appears to be less specific than PcTx1, also inhibiting rASIC3 and heterotrimeric mASIC1a/2, but with a reduced affinity (no effect on ASIC2a and ASIC2a/2b, ASIC1b not tested).

### 2.3. MitTx, a Painful Toxin

#### 2.3.1. Pharmacological Profile

MitTx was identified from the venom of Texas coral snake *Micrurus tener tener* as an α-bungarotoxin-like structure with two noncovalent subunits, a MitTx-α subunit consisting of a Kunitz type peptide of 60 amino acids, and a MitTx-β subunit, which is a 120 amino-acid phospholipase A2-like protein [99].

Independently of extracellular pH variations, MitTx was shown to constitutively activate several recombinant rodent homotrimeric and heterotrimeric ASICs [99,104], particularly rASIC1a and rASIC1b (EC_50_ = 9 and 23 nM, respectively), with a much lower effect on rASIC3 (EC_50_ = 830 nM) and on heterotrimeric rASIC1a/2a. At neutral pH, rASIC2a is not sensitive to MitTx but its proton-evoked activation is massively potentiated under more acidic conditions (pH 6.5) (Table 3, references included). Interestingly, the effect of PcTx1 and MitTx on rASIC1a were not additive [99], suggesting common binding sites. The effect of MitTx on native mouse ASIC channels in sensory trigeminal ganglion neurons, as well as the painful sensation induced by injection into the mouse hind paw, seems to mainly depend on ASIC1a-containing channels because these effects disappear in ASIC1a-KO mice.

MitTx, which locks ASICs in the open state, was co-crystallized with the cASIC1a channel to solve the first physiologically relevant open structure of these channels and to address the structure of the selectivity filter [51]. It nicely illustrates how toxins targeting ASICs are important tools not only to decode the physiological roles of these channels, but also to decrypt their structural and functional features.

#### 2.3.2. MitTx, a Gating Modifier Stabilizing the Open State

Unlike PcTx1, which locally mimics protons by targeting the acidic pocket at the interface of two ASIC subunits, MitTx interacts with a single subunit by forming extensive interactions with the wrist, palm and thumb domains of ASIC1 and acts like a ‘‘churchkey’’ bottle opener [51] (Figure 3B and Figure 5B). Several key contacts are necessary to set up this mechanism. The MitTx-α subunit insinuates the aromatic ring of its Phe14 at the interface of two channel subunits and splays them apart by forming extensive interactions with the β1-β2 linker and with the thumb domain of the adjacent subunit, both critical for the gating [12,134,135,136,137,138,139]. On the other hand, the MitTx-α subunit insinuates an ammonium group provided by its Lys16 into the wrist region, thereby coupling the base of the thumb to the TM1 domain (Figure 5B). It is interesting to note that the ammonium group occupies the same position as Cs^+^ ions in the ASIC open state [60], underscoring the role of thumb/TM1 contact in the stabilization of the open conformation of the pore. All these contacts, together with the interaction of the MitTx-β subunit with the upper part of the thumb domain, stabilize the open state of the channel where the acidic pocket is collapsed, the molecular clutch formed by linker β11–β12 is not switched, and the extracellular vestibule is extended, leading to a stabilized symmetric open pore (Figure 3B), which does not evolve towards a desensitized state as in the case when the channel is activated by protons, or in the presence of PcTx1. During the pore opening induced by MitTx, the extensive TM2-mediated intersubunit contacts, that define the occlusion of the desensitized and closed ion channel, are disrupted [51]. This stabilization of the open state can be shown on the potentiation of the ASIC2a current by MitTx, involving a drastic shifting of its pH-dependent activation curve towards less acidic pH (pH_0.5_ shifted from 3.5 to 6.0) [99]. The overlap of the binding between MitTx-β and PcTx1 on the thumb domain explains why the binding and biological activity of MitTx and PcTx1 are mutually exclusive [99,140].

### 2.4. Mambalgin

#### 2.4.1. Pharmacological Profile

Mambalgin (Mamb) is a three finger peptide toxin of fifty-seven amino acids with 3 isoforms (each differing by only one amino acid) identified from the venom of the African black mamba *Dendroaspis polylepis* (mambalgin-1 and mambalgin-2) and from the venom of the green mamba *Dendroaspis angusticeps* (mambalgin-3) [33,98]. The three Mamb isoforms display the same pharmacological properties, and will thus not be distinguished in this review.

Mamb applied at conditioning pH 7.4 was shown to inhibit cloned rodent and human ASIC1a, rASIC1b as well as other rASIC1a-containing and rASIC1b-containing heterotrimeric channels with IC_50_ ranging from 11 to 252 nM, as well as cASIC1, without any effect on rASIC2a and rASIC3 (up to 3 µM) nor hASIC2a [98] (Table 3, references included), generally expressed in both *Xenopus* oocytes and mammalian cells, neither on a variety of ligand/voltage-gated ion channels [33].

Potentiating pH-dependent effects of Mamb were also observed on hASIC1b (from *Xenopus* oocytes and mammalian cells) and rat and human ASIC1b/3 (from mammalian cells for rASIC1b/3 and *Xenopus* oocytes for hASIC1b/3), but only for sub-maximal test pH values (6.6–6.0).

Mamb drastically inhibits ASIC currents of spinal cord and hippocampal neurons [33,37,141] in good correlation with the fact that Mamb inhibits different combinations of homo- and heteromeric ASICs thought to be expressed in central neurons (i.e., ASIC1a, ASIC1a/2a and/or ASIC1a/2b). In rat sensory neurons, Mamb inhibits about 60% of ASIC mean current amplitude and PcTx1 about 40%. The difference was attributed to the additional inhibition by Mamb of ASIC1b-containing channels in addition to homotrimeric ASIC1a also inhibited by PcTx1 [33,120,142]. Accordingly, in vivo analgesic effects in rodents are in good agreement with an inhibition of ASICs, confirmed by experiments with genetic invalidation of either ASIC1a or ASIC1b [33,142,143]. Mamb also potently inhibits (by 90%) hASIC currents recorded from human stem cell-derived sensory neurons [144].

#### 2.4.2. Mambalgin, a Gating Modifier Stabilizing the Closed State

##### 
Biophysical Mechanisms and Relevance to In Vivo Analgesic Effects


On rASIC1a and hASIC1a, Mamb acts by a rightward shift of the pH-dependent activation curve towards more acidic pH values, thus stabilizing the channel closed state (Figure 4E,F) [33,34], without significant effect on the shift of the SSD curve [131]. Contrary to what was observed on rASIC1a and hASIC1a, Mamb is able to shift the pH-dependent SSD curve of rASIC1b and hASIC1b (Figure 6A,D) [34] towards more alkaline pHs, as observed with PcTx1 on rASIC1a and hASIC1a (Figure 4A,B).

Consequently, Mamb actually shows dual effects on hASIC1b and hASIC1b/3, either potentiation or inhibition, depending on both conditioning and test pH for channel activation, as illustrated by original data shown in Figure 6. Because of the shift of the activation curve towards less acidic pH (Figure 6A,B), the hASIC1b current activated from pH 7.4 to 6.0 is potentiated by Mamb. A partial inhibition is observed when the current is activated from pH 7.4 to 5.0 [34] that can be further increased when starting from a conditioning pH 6.6, because of the shift of the SSD curve towards less acidic pH (Figure 6A,C). The inhibitory effect of Mamb on rASIC1b is also pH-dependent, being stronger when the conditioning pH is slightly acidified. When the current is activated from pH 7.4 to 6.0, Mamb produces a partial inhibition (Figure 6E), whereas the current is fully inhibited upon activation by a pH drop from 6.6 to 5.0 (Figure 6F). This would support a higher potency of Mamb on both rodent and human ASIC1b in pathological situations where the extracellular pH is thought to be slightly acidified [84,145]. Like human channels [34], heterotrimeric rASIC1b/ASIC3 are weakly inhibited when the current is activated from 7.4 to 5.0 (Figure 6G,I) and can be potentiated upon activation by moderate acidosis (pH 6.6, Figure 6G,H).

These pH-dependent dual effects do not hinder the peripheral analgesic effects of Mamb in vivo against inflammatory and neuropathic pain described in rats and mice nor the participation of rASIC1b in these effects [33,142,146]. As it is clearly not the case in rodents despite similar pH-dependent effects, the dual pH-dependent effects of Mamb on hASIC1b-containing channels is therefore not expected to compromise possible analgesic effects in humans, as questioned recently [34]. In addition, analgesic effects of Mamb are not only supported by the inhibition of peripheral ASIC1b-containing channels, but also by the inhibition of ASIC1a-containing channels, as described in the central nervous system [33,142], and the rodent and human homotrimeric ASIC1a can be potently blocked by the peptide as well as heterotrimeric hASIC1a-containing channels (ASIC1a/2a, ASIC1a/1b, ASIC1a/3) [33,34].

##### 
Structural Mechanisms


Mamb interacts with rASIC1a directly through the thumb domain, but its inhibitory effect probably requires indirect influence of the palm and the β-ball domains [49]. In the structure of the hASIC1a/Mamb complex obtained by cryo-electron microscopy [52], Mamb preferentially binds to a channel conformation similar to the closed state [50,52]. In the cASIC1/Mamb complex, fingers I and II of Mamb bind to the α4 and α5 helices of the thumb domain, delimiting a part of the acidic pocket [108]. At the rat ASIC1a/Mamb interface, it was shown that the binding site is composed of four key residues forming a hinge between the α4 and α5 helices [106] (Figure 3A and Figure 5C). Mamb locks it, preventing its motion during proton activation and leading to stabilization of the expanded shape of the acidic pocket and thus to stabilization of a channel conformation similar to the closed state [50,106]. Contrary to PcTx1 and similarly to MitTx, Mamb would not interfere directly with the acidic pocket [49,105] but only on the thumb domain [106,108]. However, contrary to MitTx, the core of Mamb would have no contact with the lower part of the thumb domain [106], which could explain their different effects. Contrary to PcTx1 and GMQ effects on rASIC1a, that showed an overlap of their mechanisms, no overlap is observed with Mamb, in good agreement with the fact that this toxin does not significantly affect the desensitization process [131].

By homology with the PcTx1 mechanism which modulates the SSD, the difference in Mamb behavior between the ASIC1a and ASIC1b isoforms could be supported by the palm domain, i.e., Mamb could modify ASIC1b SSD by interfering with the palm domain of ASIC1b but not with the one of ASIC1a. This hypothesis would be interesting to explore.

Although structural models deduced from the studies of the hASIC1a/Mamb [52] and of the rASIC1a/Mamb complexes [106] are very similar, there are differences in the pharmacological behaviors depending on the species or the subunit subtypes. In the rASIC1a/Mamb interaction, the Arg28 side chain of Mamb is freely exposed to the solvent and cannot be assigned to Glu342, Asp345 or Asp349 into the acidic pocket contrary to what is suggested for the hASIC1a/Mamb interaction [52]. The direct interaction of Arg28 of Mamb with Asp351 of hASIC1a in the acidic pocket would hinder its interaction with Arg190 located in the β-ball which is mandatory for collapse of the acidic pocket [12,52]. However, Mamb was shown to still be able to partially inhibit a rASIC1a channel where Arg190 was mutated, thus disrupting the inhibitory effect of PcTx1 targeting the acidic pocket [49], which does not support a central involvement of Arg190 in the Mamb inhibition, at least in rASIC1a.

Mamb is more efficient on rASIC1a (IC_50_ = 3–55 nM, Table 3) than on hASIC1a (IC_50_ = 24–203 nM, see Table 3 for references) despite a high degree of sequence identity (98.11%, Table 1). There are two interesting differences between rat and human sequences (also true for ASIC1b), near the wrist region in the thumb domain: an insertion of two residues (Asp298 and Leu299) is found in the human sequence, as well as a Lysine at position 291 instead of Asn291 in the rat sequence. Both positions have been tested [128] and it was revealed that most of the difference of Mamb affinity between hASIC1a and rASIC1a comes from the N291K variation. Another difference could be due to the fact that, in the rASIC1a/Mamb interaction, the Lys-8 of Mamb cannot be assigned to Asp298 or Asp296, as proposed in the hASIC1a/Mamb model [52,106].

### 2.5. APETx2 and APETx-like Peptides

#### 2.5.1. Pharmacological Profile

**APETx2** is a peptide of forty-two amino acids isolated from the venom of the sea anemone *Anthopleura elegantissima* [97], belonging to the disulfide-rich all-β structural family [147]. It was shown to inhibit rASIC3 (IC_50_ of 37–63 nM), hASIC3 (IC_50_ of 175–344 nM), rASIC2b/3 (IC_50_ of 117 nM) and, at higher concentrations, rASIC1a/3 (IC_50_ of 2 µM) and rASIC1b/3 (IC_50_ of 900 nM) (Table 3, references included). APETx2 rapidly and reversibly inhibits the transient ASIC3 peak current and the sustained window current evoked at test pH 7.0 [148], as well as the alkali-induced hASIC3 current [22], but the toxin does not affect the sustained component evoked at acidic test pHs [97].

Potentiating effects of APETx2 in the micromolar range were also reported on rASIC1b, rASIC2a, rASIC1b/3 and rASIC2a/3 heterologously expressed in *Xenopus* oocytes ([75] and unpublished data). However, no effect of similar concentrations was reported on the same channels expressed in a mammalian cell line [97], highlighting differences that can occur depending of the expression system.

On native ASIC currents of rodent neurons, APETx2 was only described to exert inhibitory effects interpreted as the inhibition of ASIC3-containing channels involved in the total current [97,120,143]. APETx2 injections in rodents pain models induced anti-hyperalgesic or analgesic effects against inflammatory pain [75,91,118,149,150,151], chemical pain [152], migraine [153], joint pain [154,155], ocular pain [156], muscular pain [148,157,158] and bone pain [159,160,161]. However, inter-animal variability was reported once in a rat model of inflammatory hyperalgesia using outbred animals, with 50% of non-responsive animals proposed to be linked to the potentiating effects of APETx2 at high dose [117] on ASIC currents that could counteract the analgesic effects.

At generally higher concentrations, APETx2 was shown to inhibit recombinant and native Nav1.8 voltage-dependent channel, a sensory neuron-specific Nav channel (IC_50_ of 2.6 µM for native channel in rat sensory neurons, 55 nM for recombinant rat channel and 6.6–18.7 µM for recombinant human Nav1.8 channel [162,163]), as well as Nav1.2 (IC_50_ = 114 nM [163]), Nav1.6 currents and the cardiac hERG channel in the micromolar range [164].

**Hcr 1b-1, 2, 3, 4** are APETx-like peptides from the sea anemone *Heteractis crispa* that have different effects. Hcr 1b-1 and Hcr 1b-2 inhibit hASIC3 like APETx2, but with a lower affinity (IC_50_ of 5.5, and 15.9 μM, respectively), while Hcr 1b-2 also inhibits rASIC1a with IC_50_ 4.8 ± 0.3 μM [165,166], and Hcr 1b-3 inhibits rASIC1a and rASIC3 with IC_50_ of 5 and 17 μM, respectively. Hcr 1b-4 was found to be the first potentiator of ASIC3, simultaneously inhibiting rASIC1a at similar concentrations, with an EC_50_ of 1.53 μM and an IC_50_ of 1.25 μM, respectively. Hcr1b-2 showed an analgesic activity in vivo, significantly reducing the number of writhings in an acetic acid-induced writhing test, but promiscuous effects were also reported for Hcr1b-2 (1µM), mostly inhibiting but also potentiating Kv, Nav and Cav (T type) channels [167].

#### 2.5.2. APETx2, a Pore Blocker?

Even if APETx2 was identified in 2004 [97], its binding site and mechanism of action were poorly characterized. Only one study based entirely on docking and clustering proposes two potential binding sites on ASIC3, one on the thumb domain and the other one in the lower part of the extracellular domain (Figure 5D) [129]. Both sites remain plausible but the Arg17 of APETx2, which appears to be a critical determinant of ASIC3 inhibition, would only interact at the interface in the putative site near the bottom of the extracellular domain [110].

Inhibition by APETx2 is pH-independent (pH-dependent curves of activation and SSD are not shifted by APETx2 [128]), and this is what is expected by the APETx2 hypothetical binding site in the lower part of the palm domain, allowing occlusion of the extracellular fenestration, and suggesting a pore blocking behavior and not a gating modulation like PcTx1, MitTx and Mamb. The potentiation of rASIC1b and rASIC2a currents, as well as rASIC1b/3 and rASIC2a/3 expressed in *Xenopus* oocytes induced by concentrations of APETx2 30- to 100-fold higher than the concentration inhibiting rASIC3 (Table 3), could suggest that the second binding site located on the thumb domain might be involved in regulating its gating but with a much lower affinity.

### 2.6. Other Animal Toxins Targeting ASICs

Sea anemone toxins were shown to inhibit ASICs besides APETx2. The pi-AnmTx Ugr 9a-1 (**Ugr 9-1**) peptide, with a “twisted β–hairpin” structure, from *Urticina grebelnyi* inhibits hASIC3 expressed in *Xenopus* oocytes. It completely blocks the transient peak current with an IC_50_ around 10 µM and partially inhibits the sustained current with an IC_50_ of 1.4 µM [168]. Ugr 9-1 showed analgesic effects in two rodent models of inflammatory pain [118,168]. **PhcrTx1** from *Phymanthus crucifer* was also shown to partially inhibit native ASIC currents from rat sensory neurons (IC_50_~100 nM) but also voltage-gated K^+^ currents in the µM range. It presents an ICK scaffold and is the first member of a new structural group of sea anemone toxins [169]. The peptide **RPRFamide** from the marine cone snail *Conus textile*, related to the snail FMRFamide peptide, potentiates rASIC3 expressed in *Xenopus* oocytes, particularly the sustained component of the current (10–100 µM), therefore potentiating muscular acid-induced pain in mice in an ASIC3-dependent manner [170]. **α****-Dendrotoxin** from the green mamba *Dendroaspis angusticeps* reversibly inhibits the transient ASIC currents in rat DRG neurons with an IC_50_ of 0.8 μM, also inhibiting the sustained current at 3 μM [171]. It is a peptide of fifty-nine amino acids with a single Kunitz domain fold [172] similar to the α-subunit of MitTx (32% identity, 55% homology [173]), and is a well-known low nanomolar Kv1.x channel blocker. Finally, the short peptide **Sa12b** (10 amino acids, no cysteine) from the wasp *Sphex argentatus* was described to inhibit ASIC currents from rat sensory neurons (IC_50_ = 81 nM) when applied in the conditioning as well as in the acidic test pH. Sa12b activity would not be pH-dependent, not interacting with the proton-gating mechanism [174]. An inhibitory effect of Sa12b (1 µM) was observed on cloned rASIC1a (−38.3 ± 6.5%, n = 8, *p* = 0.019 with paired t-test, synthetic Sa12b applied in conditioning pH 7.4 and test pH 5.5, unpublished data) expressed in *Xenopus* oocytes.

## 3. Expression of ASICs in the Nervous System and Peptide Toxin Effects on Native Currents

### 3.1. Expression of ASICs in Neurons

In vivo, the effects of ASIC-targeting compounds on pathophysiological processes are thought to depend on the mixture of functional ASICs involved in native ASIC currents, depending on the expression pattern of ASIC subunits in each cell type. Although expressed in various tissues in rodents like in humans [76,81], ASICs are largely found in neurons of both the central and the peripheral nervous system (Figure 7).

#### 3.1.1. Expression in Peripheral Sensory Neurons

In rodents, almost all ASIC genes (except ASIC4) are expressed in sensory neurons of the peripheral nervous system [76,79] in the dorsal root ganglia DRG (Figure 7) [42,176,177], the trigeminal ganglion TG [178,179], and the nodose and jugular ganglia from the vagus nerve [180,181,182]. The mesencephalic trigeminal nucleus only expresses ASIC1b, ASIC2a and ASIC3 [179,183].

Among the 17 subtypes of rodent DRG sensory neurons classified by large-scale single-cell RNA sequencing (scRNAseq), ASIC genes were found to be expressed in 14 subtypes [184]. ASIC1 (without discriminating between ASIC1a and ASIC1b splice variants) is expressed in most peptidergic neurons (four subtypes over eight, including myelinated Aδ-nociceptors), in neurofilament-expressing myelinated low-threshold mechanoreceptor (LTMR) and proprioceptor neurons for which ASIC1 mRNA is a characteristic marker, and to a lesser extent in some unmyelinated non-peptidergic neurons C-LTMRs (low threshold mechanoreceptors). ASIC2 is particularly highly expressed in all unmyelinated non-peptidergic neurons (six subtypes including C-LTMR and C-nociceptors), but also in peptidergic (six PEP subtypes over eight including unmyelinated C-nociceptors and myelinated Aδ nociceptors) and neurofilament-expressing myelinated neurons (LTMR). ASIC3 is highly expressed in unmyelinated peptidergic C-nociceptors and myelinated Aδ nociceptors (for which it is a marker), but also in neurofilament-expressing myelinated neurons (including LTMRs and proprioceptors) (data from the Mouse Brain Atlas of the Linnarsson Lab., http://mousebrain.org/) [177,184,185,186] (accessed on 1 June 2022). Classification of mice TG sensory neurons based on scRNAseq identified 13 neuronal types showing great similarities with the DRG subtypes, despite the lack of proprioceptors. Among TG sensory neurons, ASIC1 is expressed in cold nociceptors, large mechanosensory touch neurons (LTMRs) and peptidergic nociceptors, ASIC2 is present in touch C-fibers, non-peptidergic and peptidergic heat nociceptors and mechanonociceptors, and ASIC3 is expressed in large mechanosensory touch neurons, large nociceptors and peptidergic heat nociceptors [187,188].

An in situ hybridization study (RNAscope) on mouse lumbar DRG neurons showed that none of the five ASIC (except ASIC4) mRNAs showed a similar distribution [189]. ASIC mRNAs, including the splice variants, were expressed in myelinated neurons (including LTMRs, proprioceptors and Aδ-nociceptors). In non-myelinated C-nociceptors, ASIC2b was expressed in almost all neurons, ASIC1a, ASIC1b and ASIC3 were only expressed in peptidergic neurons, and ASIC2a was mostly expressed in non-peptidergic neurons [143,189]. Non-peptidergic nociceptors thus showed low expression of ASIC3 and the highest levels of ASIC2a and ASIC2b expression, with ASIC1a and ASIC1b not detected, in agreement with electrophysiological, immunohistochemical, and other in situ hybridization studies [190]. This is contrasting with the peptidergic subpopulation, in which more than 60% of neurons express the ASIC3 mRNA and approximately 25% express ASIC1a and ASIC1b mRNAs [189]. An increased expression of ASICs is associated with an increase in evoked and spontaneous excitability of small size nociceptor neurons, which may contribute to hyperalgesia and chronic inflammatory pain [191,192,193].

In humans, recent scRNAseq experiments using lumbar DRGs [194] showed that nociceptors represented ~60 to 70% of all sensory neurons. Humans also have Aβ-fiber nociceptors but non-peptidergic neurons do not exist (i.e., all sensory neurons are peptidergic). Like in rodents, ASIC1 and ASIC3 are expressed in human DRG (Figure 7) and TG neurons, ASIC2 being expressed at a lower level [175,195], with little or no expression of ASIC4. ASIC3 is part of the 30 most selectively expressed ion channels in the human TG and DRG compared to the brain or other non-nervous tissues, and ASIC1 and ASIC2 were found to be enriched in neurons compared to non-neuronal cells in rat and human TG and DRG [196]. Like in mice, ASIC1 is a marker of human proprioceptors [185]. ASIC1 and ASIC3 are notably expressed in putative “silent” nociceptors in humans [194], which correspond to a subset of C-fibers specifically expressing the cholinergic receptor nicotinic alpha 3 subunit (*CHRNA3*) that innervate joints, viscera and skin and are often referred to mechano-insensitive C-fibers [197]. They are unresponsive to noxious mechanical stimuli under normal conditions, but they can be sensitized after inflammatory stimulation as they express a wide array of receptors amongst which there is ASIC3, which might be important in certain pain disorders. Comparison with gene expression in mice [184,185] shows that ASIC1 is more widely expressed among DRG subpopulations in humans than in mice although it remains enriched in Aβ-LTMRs and proprioceptors. ASIC2, in contrast, is less widely expressed in human DRGs than in mice, mainly in Aδ-nociceptors and proprioceptors, and ASIC3 is largely expressed and to a higher level than in mice, principally in pruritogen receptor enriched nociceptors, in Aδ- and Aβ-nociceptors [194].

#### 3.1.2. Expression in Central Neurons

In the rodent central nervous system, ASIC1a, ASIC2a and ASIC2b isoforms are widely expressed (Figure 7). scRNAseq experiments from mice cortex and hippocampus show that ASIC1 and ASIC2 genes are largely expressed in inhibitory and excitatory neurons in all neuronal subtypes, while ASIC3 (shown to be expressed in some brain areas) and ASIC4 are more expressed in inhibitory (GABAergic) neurons and some excitatory neurons. ASIC1 and ASIC4 were found to be characteristic markers of two distinct subclasses of inhibitory neurons in the midbrain [184,198]. The expression of ASIC1 and ASIC2 genes was shown in most inhibitory and excitatory neuronal subpopulations of the spinal cord [184,199,200] (Figure 7), in good agreement with functional electrophysiological studies on cultured neurons or dorsal spinal cord slices showing that ASIC currents were flowing through a mixture of homotrimeric ASIC1a and heterotrimeric ASIC1a/2 [37,122,123,201]. Although mainly expressed in sensory neurons, ASIC3 was also shown to be expressed in some areas of the rodent brain, particularly in pathologic states. In the hypothalamus and trigeminal nucleus caudalis, its expression is up regulated in a dural inflammatory mediated preclinical model of migraine [202], and in neuropathic mouse, ASIC3 was shown to be expressed in three brain regions (nucleus accumbens, medial prefrontal cortex and periacqueductal grey) of the pain brain network [203].

In humans, scRNAseq experiments from multiple cortical areas show that ASIC2 is widely expressed in inhibitory and excitatory neurons in all neuronal subtypes, but ASIC1 is mainly present in inhibitory neurons, whereas it is less widely expressed than in mice, and ASIC4 is likewise expressed in inhibitory (GABAergic) neurons [198]. However, contrary to rodents, the ASIC3 gene appears to be expressed in human spinal cord neurons (Sensoryomics website, https://paincenter.utdallas.edu/sensoryomics/ (accessed on 1 June 2022)) [22,175] (Figure 7).

#### 3.1.3. Expression in Glial Cells

At lower expression levels than in neurons, ASICs (ASIC1a, ASIC2a and to a lesser degree ASIC3 or ASIC4) were shown to also be expressed in rodent and human glial cells that are involved in synaptic transmission and inflammatory responses in the nervous system [204,205,206,207,208]. ASIC1a expression seems prevalent, but the LPS (lipopolysaccharide) stimulation of cultured rat microglia induces an up regulation of ASIC1a, ASIC2a and ASIC3 expression and an increase in ASIC current, leading to a subsequent increase in intracellular calcium and expression of inflammatory cytokines that could be partly inhibited by PcTx1 [205]. Similar results were obtained from mouse astrocytes along with an increase in ASIC1a expression by pharmacological induction of epilepsy that also appears in hippocampal astrocytes from epileptic patients [207]. This suggests that substantial expression of ASICs in glial cells could also possibly be related to neuronal pathological states.

### 3.2. Effects of ASIC-Targeting Peptide Toxins on Native Currents

In rodent sensory neurons, Mamb was shown to inhibit about 60% of ASIC currents and for PcTx1 about 40%, which was attributed to the supplemental inhibition by Mamb of ASIC1b-containing channels in addition to ASIC1a-containing channels, while APETx2 inhibits Mamb-insensitive ASIC3-containing channels [33,97,120,143]. In central neurons, Mamb was accordingly found to drastically inhibit the ASIC currents of spinal cord and hippocampal neurons, whereas PcTx1 was found to only inhibit about 30% of their amplitude [33,37,121], supporting a combined expression of homotrimeric ASIC1a and heterotrimeric ASIC1a/ASIC2.

Data obtained on native human ASIC currents are in good agreement with the results from rodent neurons. Two studies on cultured human DRG neurons (after therapeutic ganglionectomy on patients suffering from chronic intractable pain) show that every recorded neuron was able to generate a transient ASIC-like current upon extracellular acidification, sometimes associated with a sustained TRPV1-like current [209,210]. Another study shows that ASIC currents recorded from human stem cell-derived sensory neurons [144] were inhibited by PcTx1, APETx2 and Mamb, with the order of efficacy Mamb > APETx2 > PcTx1, suggesting the involvement in human sensory neurons of ASIC1a- and/or ASIC1b-containing heterotrimeric channels, and also of ASIC3-containing channels. Another study combining transcriptomic RNAseq and electrophysiology on DRG neurons showed a sexual dimorphism in neuropathic patients, ASIC1 and ASIC3 genes being more expressed in males than in females [211].

In cultured central cortical neurons from patients undergoing craniotomies for the removal of brain tumor, 10 nM PcTx1 was found to inhibit by 70% the amplitude of the native ASIC current [212], suggesting the major involvement of homomeric ASIC1a in human central neurons, although the ASIC2a subunit was found co-expressed with ASIC1a with a similar ratio as in mice (with ASIC2b less expressed) [213]. A higher membrane targeting of the ASIC1a subunit was observed in acutely resected human cortical tissue (from patients undergoing surgical treatment of intractable epilepsy) compared to mice, possibly linked to a more efficient trafficking due to an amino acid difference at position 285 between mASIC1a and hASIC1a [213]. A dominant expression of ASIC1 compared to ASIC2 was also reported from human central neuronal cell line cultures [214], associated with a dominant ASIC1a current (strongly inhibited by PcTx1) and a native hASIC1a current was also recorded from neuroblastoma differentiated into neuronal-like phenotype, associated with the co-expression of ASIC2 but not ASIC3, leading to some Mamb-sensitive heterotrimeric ASIC1a/2a [215].

## 4. Pathophysiological Relevance of ASICs and in Vivo Effects of ASIC-Targeting Peptide Toxins

### 4.1. Relevance in Pain

Clinical data support the involvement of ASICs in **cutaneous pain** in humans. A decrease in pH in the skin of human volunteers was associated with non-adapting pain [216] and this cutaneous acid-induced pain is blocked by amiloride and/or NSAIDs [86,87,217], with a prominent effect for 7.4 < pH < 6.0 while pain associated with more acidic pH is also sensitive to capsazepine (an inhibitor of TRPV1) [87]. The respective role of ASIC and TRPV1 channels in human cutaneous acidic pain may be complex, as a recent clinical psychophysical study on 32 healthy volunteers suggests that TRPV1 would be the predominant sensor of pH 6.0-induced pain in skin [218]. The extreme pain evoked by the Texas coral snake’s bite in humans [219] has also been attributed to the constitutive activation of human ASICs by MitTx present in the snake’s venom [99], that would be an advantage on an evolutionary point of view against mammals that represent a threat (the snake living in urbanized territories). In rodents, APETx2 has shown potent analgesic effects after local application in cutaneous acidic and inflammatory pain supporting the role of peripheral ASIC3 in primary thermal and mechanical hyperalgesia [91,117,149,152,168], and the toxin Ugr 9-1, which also inhibits ASIC3 channels, reversed inflammatory and acid-induced pain after i.v. injection [168]. Intraplantar analgesic effects of Mamb support the involvement of DRG-specific ASIC1b-containing channels in rodent pain sensing [33]. ASIC1a channels could also be involved, particularly in the orofacial region (innervated by TG neurons), where PcTx1 was shown to reduce chemically induced sub-cutaneous pain [178], whereas it was generally without effect when applied elsewhere on the body skin, perhaps due to a higher ASIC1a expression in TG neurons than in DRG neurons [220] in rodents. From an ethologic point of view, it is interesting to note that the activation of cASIC1a channels by PcTx1 would represent an evolutionary advantage for spiders to induce pain in avian species that pose a threat in their environment, whereas inducing analgesic effects in prey like rodents could be useful to prevent them for fleeing or reacting to their bite. Similarly, analgesic properties of Mamb could constitute an advantage for mamba snakes against prey (rodents).

In rodent **migraine** models, amiloride [221] and APETx2 [153,222] show analgesic effects, suggesting a role of peripheral ASIC3 in dural afferents in migraine-related behavior, whereas Mamb effects [146] also suggested a role for peripheral ASIC1, most probably ASIC1b-containing channels, particularly in the chronification of cutaneous allodynia. In a small open clinical study on migraine, amiloride showed some efficacy for the reduction of aura and headache symptoms [223,224].

Blockade of ASIC3-containing channels exerts analgesic effects in several animal models of **bone pain** [159,160,161]. Inhibiting ASIC3 attenuates pain behaviors in animal models of osteoporosis, bone cancer and osteoarthritis, and bone pathologies in which inflammation is a major component [154,159,225]. The in vivo effects of APETx2 and electrophysiological recordings of bone afferent neurons on a bone-nerve rat preparation both suggest a role for ASIC3-containing channels in Aδ and C-fiber bone afferent neurons in the pathogenesis of inflammatory bone pain [226].

Regarding **joint pain**, ASIC3 was found to be expressed in more than 30% of DRG neurons innervating the knee joint in mice [227], and ASIC expression in DRG is increased in mice models of acute arthritis or rheumatoid arthritis [227,228]. LPC and arachidonic acid (AA) were shown to induce a slow constitutive activation of ASIC3 including the human isoform [21], and high levels of lysophosphatidylcholine (LPC) were measured in synovial fluids of two independent cohorts of patients with rheumatic diseases, correlated with pain outcomes in the cohort of osteaoarthritis (OA) patients [20]. LPC also evokes a robust depolarizing current in DRG neurons at physiological pH 7.4, increases the firing of spinal nociceptive neuron innervated by nociceptive C-fiber, and induces pain behavior in rats and mice after subcutaneous co-injection with arachidonic acid, effects that are significantly reduced by ASIC3 blockers, including APETx2, or in ASIC3 knockout mice [20,21,229]. In a pathology-derived mouse model, intra-articular injections of LPC trigger a chronic pain state associated with anxiety-like behaviors that involves ASIC3-containing channels and is significantly reduced by intra-articular APETx2 [20]. APETx2 was also shown to reduce pain progression when injected in the early phase of an OA rat model [154]. This suggests a role for ASIC3 in triggering chronic joint pain, with potential implications of its inhibition for pain management in OA and possibly across other rheumatic diseases.

The role of ASICs was documented in **gastrointestinal pain** [230]. Increased hASIC3 expression in inflamed Crohn’s disease intestine and small diameter sensory neurons of the neuronal plexus suggests a role in pain or dysmotility [231]. Considering oesophageal heartburn pain associated with gastroesophageal reflux disease (GERD), ASIC1 and ASIC3 expression were recently found to be increased in biopsies of patients compared to healthy subjects, which positively correlates with symptom severity of heartburn and regurgitation [232]. In the same study, injections of PcTx1 or APETx2 were found to normalize pain response to oesophageal distension in a rat model of reflux oesophagitis, leading to visceral mechanical hypersensitivity [232]. A clinical study was conducted in 2015 in a model of GERD showing that a small molecule (PPC-5650) with a weak selectivity for ASIC1a, reduced the sensitization to mechanical stimulation of the oesophagus [233].

ASICs were also proposed to be involved in **fibromyalgia and muscle pain** [234]. ASIC3 is expressed in more than 50% of small muscle sensory afferents in rat [148,186], and ASIC expression in DRG is increased in mouse models of muscle inflammation [235]. Fibromyalgia is commonly considered as a stress-related chronic pain disorder. The involvement of ASIC3 channels in this pathology was proposed based on their potentiation by LPC [21], as excessive oxidative stress and LPC (the LPC16:0 species) levels were reported in patients with fibromyalgia [236]. Moreover, LPC level was correlated with pain symptoms in patients with high oxidative stress and disease severity, and an increase in LPC was also observed in a mouse model of stress-induced chronic hyperalgesia, in which pharmacological or genetic inhibition of ASIC3 impeded the development of chronic hyperalgesia [236]. By combining ASIC3-knockout mice and APETx2, ASIC3 was also involved in mechanical hyperalgesia in a mouse model of fibromyalgia induced by repeated intramuscular acid injections [143,157], as well as in a rat reserpine-induced pain model [237], associated with an increased expression of ASIC3 in DRG [237,238], spinal cord and thalamus [238]. Activation of ASIC1b-containing channels was involved in the mouse model of fibromyalgia induced by intramuscular acid injections, with the development of hyperalgesia absent in ASIC1b-knockout mice and blocked by Mamb but not by PcTx1 [143]. PcTx1 was, however, shown to prevent activity-induced muscular hyperalgesia in mice [116], suggesting that ASIC1a could be important in the generation of muscle inflammation [235]. APETx2 was also able to relieve pain in a rat plantar incision model of **postoperative pain** [148].

In rodents, ASIC3 was shown to contribute to **orthodontic pain** [239,240,241], and recent data show that periodontal acidification (around pH 7.0) induced by tooth movement results in mechanical tooth pain hypersensitivity that was partially reversed by an injection of APETx2 in the periodontal tissue in rats [242]. A genome-wide association study suggests that a genetic variation in the ASIC2 gene could be significantly associated with severe gingival inflammation, linked to periodontitis [243].

In addition to peripheral sensory mechanisms, ASICs were shown to be involved in **spinal and supra-spinal pain processing**. In mice, i.t. and i.c.v. injections of PcTx1 induce potent analgesic effects in inflammatory and neuropathic pain models [123], as well as in a rat model of irritable bowel syndrome [244], involving the blockade of ASIC1a which causes a release of endogenous Met-enkephalin in the cerebrospinal fluid. Central injections (i.t. or i.c.v.) of Mamb in mice also induce analgesic effects on acute, inflammatory and neuropathic pain [33,142].

### 4.2. Relevance in Other Pathological Situations

ASICs are widely expressed in neurons outside the pain pathways, as well as in various non-neuronal tissues [76,81], thus supporting their involvement in others pathophysiological processes.

Interestingly, **neuroprotective effects** could be expected from ASIC inhibition, which would be complementary to the analgesic effects, particularly when neuronal damages are associated with pain, like in neuropathic or ischemic-related pain. In rodent CNS, ASICs, and particularly ASIC1a (whose opening could induce direct and indirect Ca^2+^ entry in neurons), were shown to participate in acidotoxicity and neuronal death associated with **ischemia** or **traumatic injury** [93,245,246]. Surviving neurons after ischemia/reperfusion protocol had increased levels of ASIC2a expression, whereas ASIC1a and ASIC2b levels remained unchanged, suggesting a potential protective role of ASIC2a-containing channels [247,248]. The i.c.v. injection of amiloride or PcTx1 protects against severe focal ischemia by reducing the infarct volume by more than 50% [246,249], i.c.v. administration of Hi1a up to eight hours after stroke shows neuroprotective potency in a rat focal ischemia model [125,250], and, in rat spinal cord, i.t. injection of PcTx1 reduces the lesion volume induced by traumatic injury [251]. When i.v. injected in mice, the C5b compound, developed from molecular interaction of PcTx1 with ASIC1a, appears to cross the blood–brain barrier and shows neuroprotective ASIC1a-dependent action, rescuing cerebral ischemia damages [133]. Amiloride showed neuroprotective as well as **myeloprotective effects** in animal models of multiple sclerosis [252,253]. Accordingly, chronic brain lesions of patients with progressive multiple sclerosis show an increased expression of ASIC1 in axons, and a pilot clinical study showed that orally given amiloride could exert neuroprotective effects [254]. Three **neurodegenerative diseases** are also suggested to have ASICs involved in their etiologies, including Parkinson’s, Huntington’s and Alzheimer’s disease [255,256]. Both amiloride and PcTx1 were found to be neuroprotective in a mouse model of Parkinson’s disease [257], and the neuroprotective effect of paeoniflorin, the principal active ingredient of an anti-Parkinson’s disease traditional Chinese medicine, may involve inhibition of ASIC1a [258,259].

In **neonatal hyperbilirubinemia**, accumulation of bilirubin in the CNS results in neurotoxicity in various brain regions. A recent study showed that bilirubin potentiated the currents mediated by ASIC1a in an acidic environment and increased neuronal excitability, Ca^2+^ overload, spike firings and cell death [260]. Consistent with these results, neonatal conditioning with concurrent hyperbilirubinemia and acidosis primed long-term impairment of sensory and cognitive deficits in vivo in mice, suggesting potential benefits of ASIC inhibition [77].

ASIC inhibitors could also be useful to **relieve intervertebral disc degeneration and arthritis**. An up-regulation of ASIC1, ASIC2 and ASIC3 expression was described in the rodent and human nucleus pulposus in intervertebral disc degeneration [261,262]. Accordingly, PcTx1 reduces the acid-induced apoptosis and Ca^2+^ levels in apoptosis of endplate chondrocytes, supporting the involvement of ASIC1a [263], and PcTx1 as well as APETx2 were also shown to block the expression of acid-induced senescence-related markers in rat and human articular chondrocytes and chondrocyte cell lines [262,264]. In degenerated intervertebral discs and **osteoporosis**, extracellular acidosis induces osteoclastogenesis through intracellular Ca^2+^ rise, and both PcTx1 and specific ASIC1a siRNA significantly inhibit these events associated with bone resorption [265]. The increase in osteoclast activity not only leads to bone remodeling but is also a source of pronociceptive factors that sensitize the bone-innervating nociceptors. Recently, in a mouse model of **rheumatoid arthritis** a link between bone erosion and pain was found in a state of subclinical inflammation that could be relieved by APETx2 and ASIC3 genetic invalidation [228]. Cartilage and bone protective effect of ASIC-targeting compounds could be of therapeutic interest associated with analgesic effects against bone and joint pain. The ASIC-targeting wasp toxin Sa12b was recently found to improve the biological activity of cultured human nucleus pulposus mesenchymal stem cells isolated from patients who underwent lumbar disc herniation surgery [266] and the design of a new hydrogel containing Sa12b is now proposed for tissue engineering clinical trials to regenerate damaged nucleus pulposus in intervertebral disc [267].

Regarding **cardiovascular homeostasis**, the role of ASICs in the local vascular control is supported by the expression of ASIC3 in muscle metaboreceptors, the sensory nerves that innervate muscle arterioles and detect changes in muscle metabolism [186,268]. Inhibition of ASIC3-containing channels in sensory neurons by APETx2, amiloride and the NSAID diclofenac block the skin vasodilation response to direct pressure (pressure-induced vasodilation or PIV) in both humans and rodents [269], and a greater protein expression of ASIC3 was measured by immunoblotting in hypertensive SHR rats [270]. Genetic and pharmacological data using PcTx1 also demonstrated a role for ASIC1a in neurons in the regulation of microvascular tone and response to CO_2_ via nitric oxide production and vasodilation [271]. In addition, ASIC1a was recently identified as a critical determinant in **heart ischemia reperfusion injury** through human genetics studies, human stem cell-derived cardiomyocytes and mouse models, confirming its potency in multiple ischemia and stroke injury models [125,127,246]. The treatment with Hi1a or PcTx1 reduces human stem cell-derived cardiomyocytes death by half after in vitro ischemia-reperfusion, and in mice, the genetic ablation or the pharmacological blockade of ASIC1a improves cardiomyocyte recovery after acute ischemia-reperfusion injury without affecting heart functional homeostasis [127].

ASIC1a channel inhibitors (PcTx1 and amiloride) cause a significant reduction of **tumor growth** and tumor load in mice [272]. Expression of ASIC1a was reported to be high in cancer patients, in vitro experiments revealed that PcTx1 or ASIC1a siRNA could weaken the migration, proliferation and invasion of tumor cells, and PcTx1 (i.v.) could inhibit breast tumor growth in mice [273]. Mamb was shown to also be efficient against growth and migration of glioma, leukemia and melanoma cells [274,275,276].

**Anxiolytic-like effects** of PcTx1 have also been described, which would be again complementary to the analgesic effects. A genetic variation of the human gene coding for ASIC1a was associated with panic disorder and with anxiety phenotypes linked to amygdala dysfunction [277,278]. Genetic disruption of ASIC1a in neurons or i.c.v. injection of PcTx1 was shown to have antidepressant effects in mice [279,280]. However, other data show that activation of ASIC1a channels in the rat basolateral amygdala decreases anxiety-like behavior, while inhibition by PcTx1 would increase the level of anxiety in rats [281]. These discrepancies could rely on activity of ASICs in other brain regions than the amygdala (like hippocampus) or on differences in ASIC1a contribution in innate and acquired fear [77].

ASIC involvement in **epilepsy** is also still unclear, with some in vivo rodent data suggesting that ASIC inhibition (particularly by i.c.v. PcTx1) can reduce epileptic symptoms [77], while other data indicated that ASICs, and especially ASIC1a, could play a role in seizure termination through activation of inhibitory interneurons when brain pH decreases [282]. In humans, genetic study suggested an association between single nucleotide polymorphisms in ASIC1 and temporal lobe epilepsy [283]. A down regulation of the expression of ASIC1a was reported in cortical lesions of patients with focal cortical dysplasia, a recognized cause of medically intractable epilepsy [284], which suggests that ASIC1a loss may contribute to epileptogenesis in these patients. An anti-epileptic role has also been suggested for ASIC3-containing channels expressed in interneurons, associated with elevated expression in the brains of temporal lobe epilepsy patients and rats, and inhibition of ASIC3-containing channels with APETx2 in rat models of epilepsy increases seizures susceptibility [285]. In addition to neuronal ASICs, an increased ASIC1a expression also occurs in mouse astrocytes after induction of epilepsy, as well as in hippocampal astrocytes from epileptic patients [207].

## 5. Conclusions

Toxins have not only been instrumental in the study of ASICs to understand the molecular features of these channels and their pH-dependent gating, but also to study the physiological and pathological roles of native channels both in vitro and in vivo in rodent pain models, supporting ASICs as therapeutic targets in pain and beyond. Furthermore, the molecular knowledge of ASIC gating and interaction with ASIC-targeting toxin inhibitors now allows to design new molecules like C5b and to predict their pharmacological potential and possibly their therapeutic relevance. Other putative applications of these peptides are also emerging, such as the design of a fusion protein incorporating an alpaca-derived nanobody targeting hASIC1a and the peptide toxin PcTx1 to achieve potent and, contrary to PcTx1 alone, more stable inhibition of ASIC1a (~84–87% current inhibition), improving the potency of PcTx1 and its potential applications [16].

ASICs exhibit complex and highly pH-dependent gating properties, and it is therefore not surprising that ASIC-targeting peptide toxins interact with channel gating also by a complex and pH-dependent mechanism. However, despite sometimes complex behaviors with pH-dependent activating, potentiating or inhibitory effects on rodent and human cloned channels recorded in vitro, the effects of ASIC-targeting toxins described on neuronal native ASIC currents in rodent and, when possible, in human neurons were mostly inhibitory for PcTx1, APETx2 and Mamb, or stimulatory for MitTx, in good agreement with in vivo effects. Insights from rodent and human studies on various pain-related processes [78] show that the ASIC-inhibiting toxins PcTx1, Mamb and APETx2 always induce a reduction of acute or pathological pain, whereas MitTx increased pain-related behaviors, which is consistent with the effects of other ASIC inhibitors like amiloride or NSAIDs, and with the effects of ASIC genetic invalidation or knockdown. Dual pH-dependent effects of the ASIC-targeting compounds do not therefore compromise their analgesic relevance both in the central and peripheral nervous system. Furthermore, complementary effects of some ASIC-targeting analgesic toxins like neuroprotective and anxiolytic effects could even be beneficial.

ASICs are thus interesting potential drug targets regarding the need to develop new and more effective analgesics with limited adverse side effects, notably in the context of the opioid crisis [286]. In addition, several administration pathways could be used to alternatively target local, peripheral or central pain mechanisms, depending on the ASIC subtype targeted, the physicochemical properties of the compounds and their blood–brain barrier permeability. Some of the ASIC peptide blockers described here could be interesting potential leads for pain relief, including in the context of chronic and inflammatory pain, with relevance in migraine, bone, joint and muscle pain, fibromyalgia, and postoperative, gastrointestinal or tooth pain, thus deserving further characterization of their effect on native ASICs in human neurons.

## Figures and Tables

**Figure 1 toxins-14-00709-f001:**
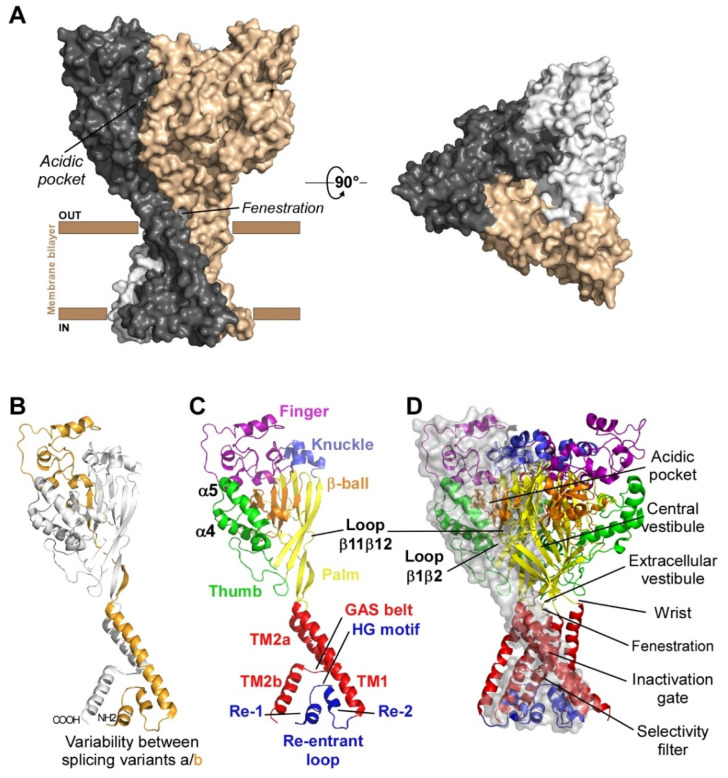
**Structure of ASICs.** (**A**) Trimeric organization of ASICs (left panel: side view, right panel: top view). (**B**) Tridimensional skeletal model of a single subunit where variable regions between isoforms “a” and “b” of rat ASIC1 and ASIC2 are highlighted in gold. (**C**) Structure of a single subunit of chicken ASIC1 in resting state (the different sub-domains are shown in specific colors; PDB ID: 6vtl). (**D**) Skeletal 3D representation of a functional channel formed by the assembly of three subunits. A transparent grey surface was added to one subunit to delineate the interface between two adjacent subunits. Same colors as in (**C**) for the different sub-domains, and key structural domains mentioned on the right. Cytoplasmic N- and C-termini, whose structures are unknown, are not shown. Designed with PyMOL software.

**Figure 2 toxins-14-00709-f002:**
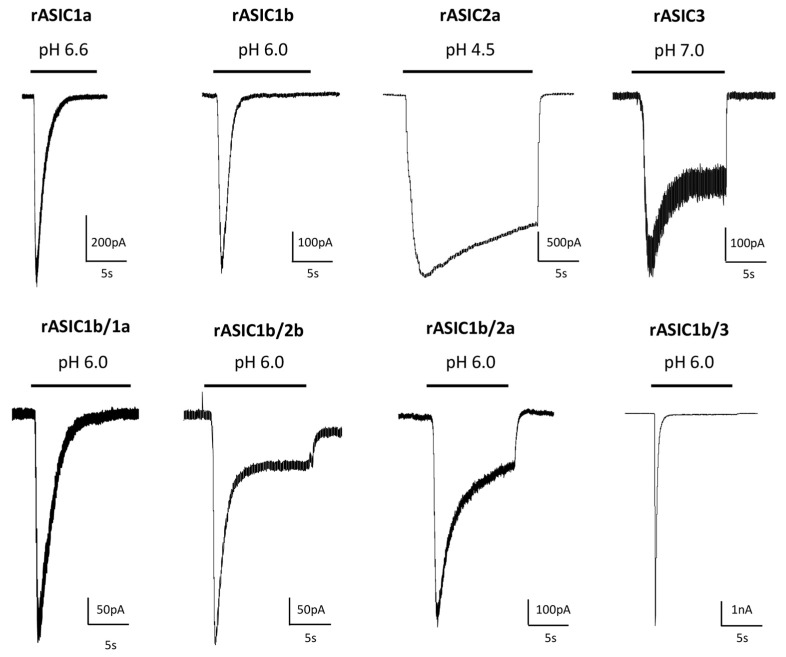
**Diversity of currents flowing through homo- and heterotrimeric cloned ASICs.** Original current traces of rat heterologously expressed ASIC currents recorded from HEK293 cells depending on the composition in ASIC subunits, activated from pH 7.4 to the indicated test pH, at −60 mV. Homotrimeric channels result from the expression of only one type of ASIC subunit (indicated above each current), whereas heterotrimeric channels result from the co-expression of two different subunits (1:1 ratio in transfection). The corresponding current noted rASIC1b/1a, for example, results from the co-expression of rASIC1b and rASIC1a subunits.

**Figure 3 toxins-14-00709-f003:**
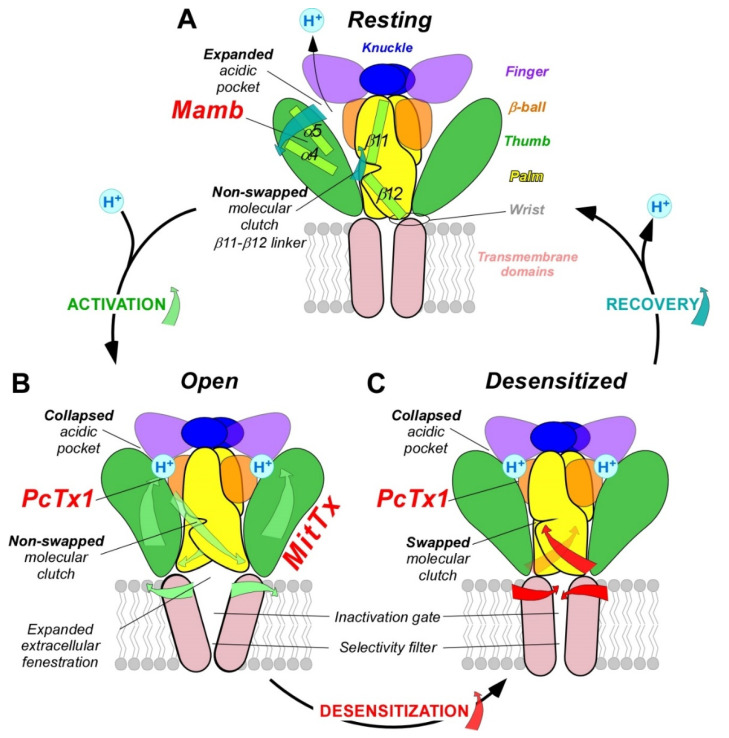
**pH-dependent gating mechanisms of ASICs and interaction with toxins.** ASIC gating involves three conformational states. (**A**), high pH resting state, which is stabilized by the toxin mambalgin (see Section 2.4) (major domains involved are indicated with the same color code as in Figure 1). (**B**), low pH open state, which is stabilized by the toxin MitTx (see Section 2.3) and also partially by the toxin PcTx1 (see Section 2.2). (**C**), low pH desensitized state also promoted by PcTx1. To illustrate the recovery process in (**A**), the deprotonation mechanism of only one acidic pocket is presented. Blue (**A**), green (**B**) and red (**C**) arrows show critical conformational changes during recovery, activation and desensitization processes, respectively. For clarity, only two subunits are shown.

**Figure 4 toxins-14-00709-f004:**
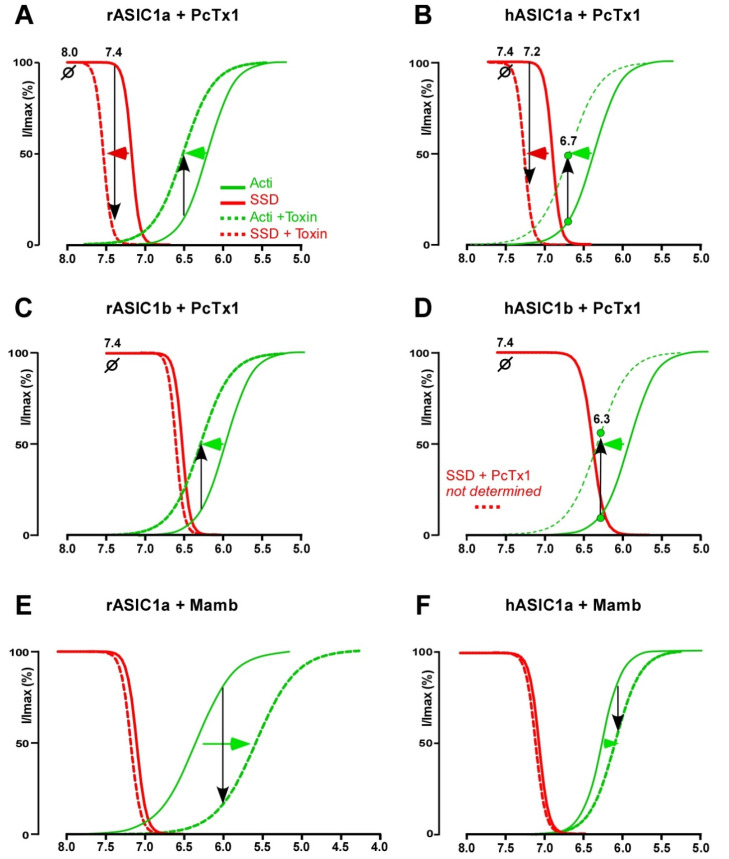
**Toxins interfere with the pH-dependent gating of rodent and human ASICs.** Schematic pH-dependent curves of normalized (I/I_max_%) activation (Acti, green) and steady state desensitization (SSD, red) of heterologously expressed cloned rat and human ASIC1a and ASIC1b in the absence (solid line) and in the presence of PcTx1 (dashed line) (**A**–**D**) and of rat and human ASIC1a in the presence and in the absence of mambalgin (Mamb) (**E**,**F**). All curves were adapted from published data. (**A**,**B**), **rASIC1a** (**A**) **and hASIC1a** (**B**) **gating modulation by PcTx1**. The toxin increases the apparent H^+^ affinity of rASIC1a current thus inducing a leftward shift of both the activation and SSD curves towards more alkaline pH values. (**A**), PcTx1 inhibitory effect on rASIC1a current from physiological conditioning pH 7.4 (black downward arrow) to every test pH is mostly due to its pH-dependent SSD promoting effect, whereas no more inhibition was observed from pH 8.0 (⊘) instead revealing a potentiation of the current at test pH in the activation curve pH range (7.2–6.2), due to the opposite potentiating effect by a leftward shift of the activation curve (black upward arrow) [29] (PcTx1 10 nM). (**B**), PcTx1 exerts almost no effect on the hASIC1a current maximally activated from conditioning pH 7.4 (⊘), an inhibitory effect on the hASIC1a current maximally activated from conditioning pH 7.2 (black downward arrow), and a potentiation on the hASIC1a current submaximally activated from conditioning pH 7.4 (black upward arrow). Curves adapted from [30,31] (PcTx1 1 nM), with the shift of activation curve deduced from the PcTx1-induced current potentiation at test pH 6.7 (green points, PcTx1 60 nM) [19]. (**C**,**D**)**, rASIC1b** (**C**) **and hASIC1b** (**D**) **gating modulation by PcTx1**. The toxin promotes opening of rASIC1b and hASIC1b through a leftward shift of the activation curve towards less acidic pH with almost no effect on the SSD curve. Consequently, PcTx1 does no inhibit the current maximally activated from conditioning pH 7.4 (⊘), and potentiates the current submaximally activated from pH 7.4 to test pH 6.8–5.8 (black upward arrow). Curves adapted from [32] (PcTx1 100 nM), and the shift of hASIC1b activation curve is deduced from the PcTx1-induced potentiation of the current at test pH 6.3 (green points, PcTx1 60 nM) [19]. The effect of PcTx1 on hASIC1b SSD curve is not yet known. (**E**,**F**)**, rASIC1a** (**E**) **and hASIC1a** (**F**) **gating modulation by Mamb**. Mamb inhibits rASIC1a and hASIC1a currents mainly by a rightward shift of the pH-dependent activation curve towards more acidic pH values. Curves for rASIC1a current adapted from [33] (Mamb-1, 200 nM), and for hASIC1a from [34] (Mamb-3, 10 nM; note that this concentration is below IC_50_ value of Mamb on hASIC1a (see Table 3) and that a higher shift could thus be expected with a higher Mamb concentration). Red and green arrows illustrate shifts (acidic rightward, alkaline leftward) in the pH-dependent curves of activation and/or SSD by toxins. Data on the gating modulation of hASIC1b and rASIC1b by mambalgin-1 are shown in another following figure.

**Figure 5 toxins-14-00709-f005:**
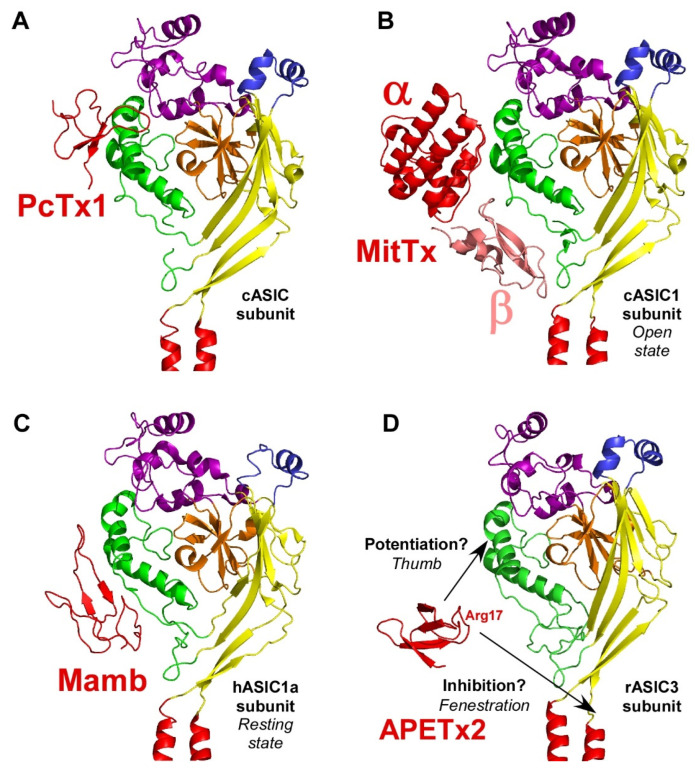
**Toxin binding sites on one ASIC subunit.** (**A**) Structure of a single cASIC1 subunit (rotated view of the skeletal 3D representation shown in Figure 1C) in complex with PcTx1 (PDB ID: 3s3x) [51]. (**B**), Structure of a single cASIC1 subunit in complex with MitTx (heterodimeric complex of MitTx-α and MitTx-β (PDB ID: 4NTY) [60]. (**C**), Cryo-EM structure of a single hASIC1a subunit in complex with mambalgin-1 (Mamb) at pH 8.0 (PDB ID: 7CFT) [52]. (**D**), Model of a single rASIC3 subunit extrapolated from cASIC1 structure, along with APETx2 (PDB ID: 2MUB) at the same scale, with two potential binding sites (black arrows) [129]. Designed with PyMOL software.

**Figure 6 toxins-14-00709-f006:**
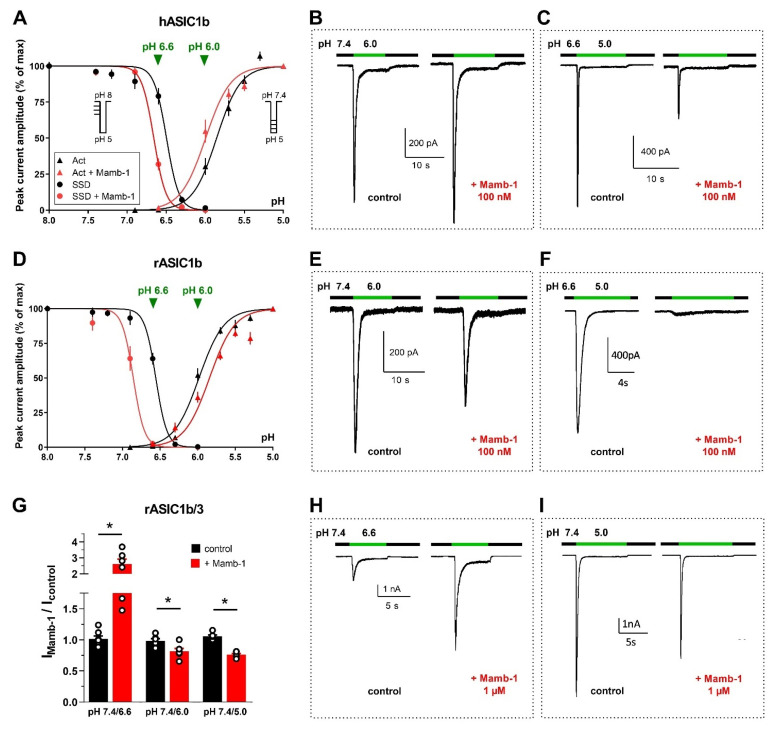
Dual pH-dependent effects of mambalgin-1 on whole-cell currents flowing through hASIC1b (**A**–**C**), rASIC1b (**D**–**F**) and rASIC1b/3 (**G**–**I**) heterologously expressed in HEK293 cells (unpublished data). (**A**) on hASIC1b current, mambalgin-1 (Mamb-1, 100 nM) induced a leftward shift of both the pH-dependent activation (Act) curve (pH_0.5_ shifted from 5.86 to 6.0, *p* = 0.02, Mann–Whitney test) supporting a potentiating effect, and of the pH-dependent SSD curve (pH_0.5_ shifted from 6.50 to 6.66, *p* = 0.004) supporting an inhibitory effect. The protocols used for activation and SSD are shown in inset in A (mean ± SEM; n = 4–12 cells per point). (**B**,**C**), original hASIC1b whole-cell current traces recorded at −60 mV illustrating the dual effects of Mamb-1 (100 nM, applied 30 s before the pH drop) on a current activated by a pH drop from 7.4 to 6.0 (potentiation, B), and on a current activated by a pH drop from 6.6 to 5.0 (inhibition, C). (**D**) on rASIC1b current, Mamb-1 (100 nM) induced a rightward shift of the pH-dependent activation curve (pH_0.5_ shifted from 5.98 to 5.85, *p* = 0.002) and a leftward shift of the pH-dependent SSD curve (pH_0.5_ shifted from 6.56 to 6.86, *p* = 0.0002), both supporting an inhibitory effect (same protocols and curve labels as in A; mean ± SEM; n = 4–12 cells per point). (**E**,**F**), original rASIC1b whole-cell current traces recorded in the same conditions as in B-C and illustrating the partial inhibition by Mamb-1 of a current activated by a pH drop from 7.4 to 6.0 (**E**), and the full inhibition of a current activated by a pH drop from 6.6 to 5.0 (**F**). (**G**) bar graph quantification with individual data points of the effects of Mamb-1 (1 µM) on rASIC1b/3 heteromeric current. Mamb-1 induced a potentiation of the current when activated by a pH drop from 7.4 to 6.6 (*p* = 0.02, Wilcoxon paired test), but a partial inhibition when activated by a pH drop from 7.4 to 6.0 (*p* = 0.03) or 5.0 (*p* = 0.03). Mean ± SEM; n = 6–7 cells per condition. * *p* < 0.05. (**H**,**I**), original rASIC1b/3 whole-cell current traces recorded in the same conditions as in (**B**,**C**) and illustrating the potentiating effect (**H**) and the partial inhibition (**I**) by Mamb-1 depending on the test pH value.

**Figure 7 toxins-14-00709-f007:**
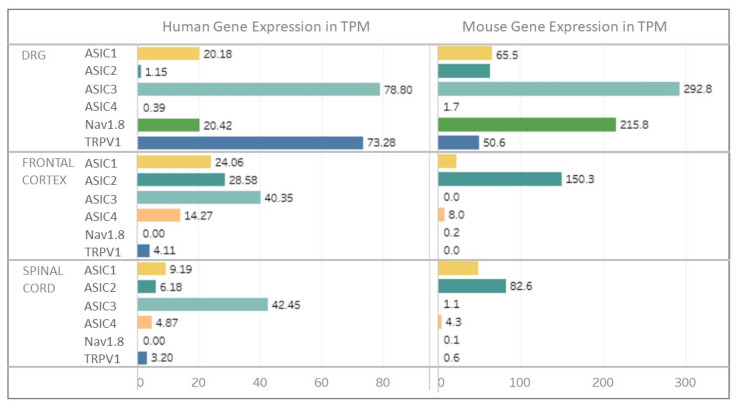
**Expression of ASICs in rodent and human nervous tissues.** Expression of ASIC genes in central and peripheral nervous system from RNAseq data for human and mouse ASIC1-4, Nav 1.8 and TRPV1 genes from dorsal root ganglia (DRG), frontal cortex and spinal cord. Nav 1.8 and TRPV1 channels were shown as typically expressed in nociceptor sensory neurons. X axis represents TPM: Transcripts Per Million. Adapted from website: https://sensoryomics.shinyapps.io/RNA-Data/ [175] (accessed on June 2022).

**Table 1 toxins-14-00709-t001:** **Protein sequence comparison of rat and human ASIC subunits**.

Isoform	Species	% Identity	Size (aa)	Name in Genbank	Sequence ID
ASIC1a	*Rattus norvegicus*	98.11%	526	ASIC1	NP_077068.1
	*Homo sapiens*		528	ASIC1 isoform b	NP_001086.2
ASIC1b	*Rattus norvegicus*	93.24%	559	ASIC1 isoform X5	XP_006257440.1
	*Homo sapiens*		562	ASIC1 isoform c	NP_001243759.1
ASIC2a	*Rattus norvegicus*	99.02%	512	ASIC2 isoform MDEG1	NP_001029186.1
	*Homo sapiens*		512	ASIC2 isoform MDEG1	NP_001085.2
ASIC2b	*Rattus norvegicus*	98.83%	563	ASIC2 isoform MDEG2	NP_037024.2
	*Homo sapiens*		563	ASIC2 isoform MDEG2	NP_899233.1
ASIC3	*Rattus norvegicus*	83.68%	533	ASIC3	NP_775158.1
	*Homo sapiens*		531	ASIC3 isoform a	NP_004760.1
ASIC4	*Rattus norvegicus*	97.22%	539	ASIC4	NP_071570.2
	*Homo sapiens*		539	ASIC4 isoform 1	NP_061144.4

Percentages of amino acid (aa) identity were calculated using BLAST.

**Table 2 toxins-14-00709-t002:** **Functional pH ranges of currents flowing through cloned rodent and human ASICs**.

Cloned Channel	ACTIVATION		SSD	
	Test pH Threshold/max	pH_0.5_	Conditioning pH Threshold/max	pH_0.5_
rASIC1a	7.0/5.5	6.4–5.8 ^chimnqtwxyz^	7.4/6.8	7.3–7.1 ^cehimtyz^
rASIC1b	6.4/5.6	6.3–5.7 ^fitwxy#^	7.3/6.6	7.0–6.5 ^fit#^
m/rASIC2a	6.0/3.0	5.0–3.8 ^bnqwxz^	7.0/4.5	6.3–5.6 ^mz^
m/rASIC3	7.2/5.5	6.8–6.3 ^otwy^	7.4/6.8	7.2–7.0 ^sty^
rASIC1a/2a	6.3/4.5	5.6–4.8 ^nqrw^		
m/rASIC1a/2b	6.8/6.0	6.4–6.2 ^pw^	7.4/7.1	7.3 ^p^
rASIC1a/1b		6.3–5.8 ^w^		
rASIC1a/3	7.0/5.5	6.7–6.3 ^rtw^	7.0/6.8	7.1 ^t^
rASIC1b/3	6.6/5.9	6.7–6.2 ^tw^	6.9/6.6	6.8 ^t^
rASIC1b/2a		4.9 ^w^		
rASIC2a/3	7.2/4.5	6.1–5.6 ^rw^		
m/rASIC2a/2b		4.8 ^bw^		
rASIC2b/3		6.5 ^w^		
m/rASIC1a/2a/3		6.4–5.1 ^rw^		
rASIC1a/2b/3		6.3 ^w^		
rASIC1b/2a/3		4.9 ^w^		
hASIC1a	6.8/6.0	6.6–6.3 ^dgikov^	7.0/6.7	7.2–6.9 ^degiko^
hASIC1b	6.5/5.5	5.9–5.7 ^gi^	6.7/6.4	6.5–6.1 ^gi^
hASIC2a	6.8/3.5	5.7 ^u^	6.0/4.7	5.5 ^u^
hASIC3a	7.0/5.5	6.6–6.2 ^aj^	7.0/7.9	7.7–7.5 ^as^
cASIC1	6.8/6.3	6.6 ^l^	7.4/7.1	7.3 ^l^

Representative pH ranges (threshold/max) and pH_0.5_ values for pH-dependent activation of cellular ASIC currents activated from conditioning pH 7.4 to variable test pHs, and for pH-dependent steady state desensitization (SSD) of currents maximally activated from variable conditioning pHs with rat (r), mouse (m), chicken (c) and human (h) homotrimeric and heterotrimeric ASICs heterologously expressed in *Xenopus* oocytes or mammalian cell lines. The corresponding current noted rASIC1a/2a, for example, results from the co-expression of rASIC1a and rASIC2a subunits. References: a [27], b [28], c [29], d [30], e [31], f [32], g [19], h [33], i [34], j [22], k [35], l [36], m [37], n [38], o [39], p [40], q [41], r [42], s [43], t [44], u [45], v [46], w [14], x [47], y [48], z [49], # unpublished data.

**Table 3 toxins-14-00709-t003:** **Effects of ASIC-targeting animal toxins on the amplitude of cloned rodent and human ASIC peak whole-cell currents activated from physiological pH 7.4**.

Channel	Mamb (1, 2 or 3) IC_50_/EC_50_	PcTx1 IC_50_/EC_50_	APETx2 IC_50_/EC_50_	MitTx EC_50_
**rASIC1a**	**INH**^🔾◆^ 3–55 nM ^a,p,q,r^	**INH** ^🔾◆^ 0.3–3.7 nM ^b,c,o,u^	**NO**^🔾◆^ at 10 µM ^y,e^	**ACT** ^🔾^ 9 nM ^i^
**rASIC1b**	**INH** ^🔾◆^ 22–192 nM ^a,p,q,r^	**POT** ^🔾^ 25–100 nM ^d, u^	**NO**^◆^ at 3 µM ^e^ **POT**^🔾^ at 3–10 µM ^y,#^	**ACT**^🔾^ 23 nM ^i^
**rASIC2a**	**NO**^🔾◆^ at 3 µM ^a,p^	**NO**^🔾^ at 100 nM ^b^	**NO**^🔾◆^ at 3 µM ^e,#^ **POT**^🔾^ at 10 µM ^y^	** POT ** ^🔾^ at 75 nM ^i^
**rASIC3**	**NO**^🔾◆^ at 3 µM ^a,p^	**NO**^🔾^ at 100 nM ^b^	** INH ** ^🔾◆^ 37–63 nM, test pH 6 ^e,f,g^	** ACT ** ^🔾^ 830 nM ^i^
**rASIC1a/2a**	**INH**^🔾◆^ 152–252 nM ^a,p,r^	**NO**^🔾◆^ at 50 nM ^b,v,w^		** ACT ** ^🔾^ at 75 nM ^i^
**rASIC1a/2b**	** INH ** ^◆^ 61 nM ^a^	**INH**^🔾^ 3 nM ^h^ **NO**^◆^ at 300 nM ^#^		
**rASIC1a/1b**	** INH ** ^◆^ 72 nM ^a^			
**rASIC1a/3**	**NO**^◆^ at 2 µM ^a,p^	**NO**^◆^ at 10 nM ^b^ **INH**^◆^ at 100 nM, test pH 6 ^x^	** INH ** ^🔾◆^ 2 µM ^e,#^	**NO**^🔾^ at 75 nM ^i^
**rASIC1b/3**	**INH**^◆^ at 1 µM ^#^ **POT**^◆^ at 1 µM, test pH 6.6 ^#^		** INH ** ^◆^ 900 nM, test pH 6 ^e^ **POT** ^🔾^ at 3 µM, test pH 6 ^#^	
**rASIC1b/2a**	** INH ** ^◆^ at 1 µM ^#^			
**rASIC1b/2b**	** INH ** ^◆^ at 100 nM ^#^			
**rASIC2a/3**			**NO**^◆^ at 3 µM ^e^ **POT**^🔾^ at 3 µM, test pH 6 ^#^	**NO**^🔾^ at 75 nM ^i^
**rASIC2b/3**			** INH ** ^🔾◆^ 117 nM ^e,#^	
**mASIC1a/2a/3**		**NO**^◆^ at 100 nM ^x^		
**hASIC1a**	** INH ** ^🔾◆^ 24–203 nM ^a,q,s,t^	**NO**^🔾◆^ at 25–60 nM ^m,n,j^ **INH**^🔾^ 3.3 nM, test pH 6 ^u^ **POT**^🔾^ at 60 nM, test pH 6.7 ^n^		
**hASIC1b**	**INH**^🔾^ at 1 µM **POT**^🔾◆^ 60 nM, test pH6 ^q,#^	** POT ** ^🔾^ at 60 nM ^n^		
**hASIC2a**	**NO**^◆^ at 850 nM ^a^	**NO**^◆^ at 100 nM ^j^		
**hASIC3a**			**INH**^🔾◆^ 175–344 nM ^e,z^	
**hASIC1a/2a**	** INH ** ^◆^ 220 nM ^a^	**NO**^◆^ at 100 nM ^j^ **POT**^🔾^ at 200 nM ^h^		
**hASIC1a/1b**	** INH ** ^🔾^ 256 nM ^q^			
**hASIC1a/3**	** INH ** ^🔾^ 462 nM ^q^			
**hASIC1b/3**	**INH**^🔾^ at 1 µM ^q^ **POT**^🔾^ at 1 µM, test pH 6 ^q^			
**cASIC1**	** INH ** ^◆^ 124 nM, test pH 6 ^t^	** ACT ** ^🔾◆^ 189 nM ^l,k,p^ ** POT ** ^🔾◆^ at 20 nM ^k,m^		

Rat (r), mouse (m), chicken (c) and human (h) homotrimeric and heterotrimeric ASICs were heterologously expressed in *Xenopus* oocytes (🔾) or mammalian cell lines (◆) and activated from pH 7.4 by an acidic maximal stimulation to pH 5.0 or below (unless another test pH is mentioned). The corresponding current noted rASIC1a/2a, for example, results from the co-expression of rASIC1a and rASIC2a subunits. Inhibition (INH) or potentiation (POT) of the H^+^-activated ASIC peak current, or activation (ACT) in the absence of acid stimulation is shown in red, blue and purple, respectively. The IC_50_ or EC_50_ values are indicated when available. If not, the highest dose tested is noted; NO: no effect (with the highest dose tested indicated). Toxins were applied at physiological conditioning pH 7.4. References: a [33], b [96], c [29], d [32], e [97], f [109], g [110], h [40], i [99], j [111], k [112], l [60], m [30], n [19], o [100,101], p [98], q [34], r [105], s [113], t [108], u [31], v [114], w [115]; x [116], y [117], z [118], # unpublished data.

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
