# Peer review of "Mechanisms of Action of the Peptide Toxins Targeting Human and Rodent Acid-Sensing Ion Channels and Relevance to Their In Vivo Analgesic Effects"

_toxins, 2022, doi:10.3390/toxins14100709_

Round 1
Reviewer 1 Report
This is a very well written and carefully organized comprehensive review on toxins that modulate acid-sensing ion channels (ASICs), which should be of relevance and interest for the readers of this journal.
There are several sematic and editorial issues that should be addressed:
1. Legend to Table 3: symbols are lacking in the brackets.
Symbols are also lacking in lns. 288, 302
2. Legend to Figure 5 and ln. 451: Font size, bold.
3. lns. 149, 170, 350 and others: -ball, -TRTX; something is lacking here
4. ln. 210: better write pathophysiological (as in ln. 214), not physiopathological
Author Response
We thank the reviewer for his comments on our manuscript.
Answer to comments 1-4 - All points have been modified as suggested. It should be noted that most semantic and editorial issues are the consequence of errors introduced during the journal formating process to obtain the version for reviewers, which modified or suppressed some symbols and special characters without the possibility for the authors to check the final version that has been sent to reviewers. We have identified and corrected several other editing problems, among which problems with a lot of bibliographic references that were no longer usable with EndNote, requiring to process them individually.
Reviewer 2 Report
The manuscript provided a review regarding to mechanisms and efficacies of peptide toxin targeting ASICs. The manuscript overall was well written in a scientific manner with several nice schemes, which could be suitable to be published in Toxins
Author Response
We thank the reviewer for his kind review.
Reviewer 3 Report
This review addresses the mechanisms by which peptide toxins modulate ASICs channels activity and thus it nicely fit the scope of Toxins. The MS is very comprehensive, covering structural, biophysical, pharmacological and physiological aspects. Molecular rendering of ASIC and toxins, as well as the scheme summarizing the gating cycle are also beautiful. Overall, the writing is clear and should be accessible to a broad audience including non-specialists and undergraduate students. Some suggestions for further improvements are provided below.
1) The review lacks an introductory paragraph introducing the topic (ASICs channels and pH-dependent gating, nociception, toxins as possible pain-relieving therapeutic candidates) from a broader perspective and defining the intended scope (and limitations) of this review. This will help catching the attention of readers beyond the specific ASIC channel field. Here, the authors should consider mentioning also other channels which show pH-dependent gating, like TRPV1 which can also be activated by proton binding (DOI 10.1038/emboj.2011.19) and CNG channels which display a pH-dependent desensitization somewhat reminiscent to ASIC (DOI 10.1113/jphysiol.2014.284216, DOI: 10.1007/s00424-021-02610-6). Interestingly, these channels are evolutionary unrelated to ASIC and belong to the voltage gated superfamily (DOI: 10.1007/s00424-021-02610-6).
2) In the introduction it should also be briefly mentioned the ionic selectivity of ASICs, which currently is not
3) Line 63, please consider defining what “selectivity filter” is to guide non-specialist readers
4) Line 67: repetition of "acid identity"..."The lowest amino acid identity is 52% amino acid identity between cASIC1 and....", please rephrase to remove redundancy
5) Figure 1, panel D: please consider rendering in color only one subunit to highlight the domain swapped architecture (two subunits can be in two different gray shades, while the third subunit, equivalent to subunit shown in panel C shall be kept with the current color scheme)
6) Line 301; please define (r), (m), (c), (h) abbreviations the first time they do appear (line 57 for cASIC1 etc…)
7) Fig. 1 and 5. Please specify in the figure legend the software used for the molecular rendering
8) Figure 6: Does the data shown in Fig. 6 come from other published works or are these original data belonging to the authors? please clarify
9) Figure 7: Please define x axis units
Author Response
We thank the reviewer for his comments and helpful suggestions to improve our manuscript.
Answer to point 1: A short introductory paragraph has been added before Chapter 1, page 1, to give the general context and scope of the review, mentioning as suggested other channels with pH-dependent gating. Some of these channels are further discussed at the end of Chapter 1.3, page 9, to, briefly compared (an extensive comparison would probably be behind the scope of this review) the molecular mechanisms of their pH-sensitivity/gating with the pH-dependent gating of ASICs.
Answer to point 2: The ion selectivity of ASICs is now mentioned in Chapter 1.2, page 4: « They are sodium selective, with additional low calcium permeability for ASIC1a and hASIC1b {Waldmann, 1997 #164}{Bassler, 2001 #91}{Hoagland, 2010 #1069} ».
Answer to point 3: The term « selectivity filter » is now explained page 4: « i.e., the structural element in the narrowest part of the pore that determines ionic selectivity”.
Answer to point 4: Modified as suggested.
Answer to point 5: Panel D of Figure 1 has been modified to show, in addition to the colored subunit domains, the surface of the same subunit shown in C as a transparent grey shading. This allows a better representation of the interface between the thumb (green) and palm (yellow) domains of two adjacent subunits. We tried also to figure two subunits in grey shades, as suggested by the reviewer, but the result was visually confusing.
Answer to point 6 : Abbreviations are defined upon first appearance.
Answer to point 7 : the PyMol software was used and is now cited in the legends of Figures 1 and 5.
Answer to point 8: The data shown in Figure 6 are original data from the authors, and « unpublished data » is now stated in the legend.
Answer to point 9: X axis represents gene expression in TPM: Transcripts Per Million. This is written at the top of the figure, and now better explained in the legend.
General remark: It should be noted that errors have been introduced during the journal formatting process to obtain the version for reviewers, with modification or suppression of some symbols and special characters without the possibility for the authors to check the final version that has been sent to reviewers. We have identified and corrected these editing problems, which explains the number of additional small modifications appearing in the revised version.
Round 2
Reviewer 3 Report
The authors have satisfactorily adderssed all my comments. Congratulations on the very nice and comprehensive review!
AM